# ROBOTARENA ∞ : SCALABLE ROBOT BENCHMARKING VIA REAL-TO-SIM TRANSLATION

**Yash Jangir**[1], **Yidi Zhang**[1]*, **Kashu Yamazaki**[1]*, **Chenyu Zhang**[1]*,
**Kuan-Hsun Tu**[2], **Tsung-Wei Ke**[2], **Lei Ke**[1], **Yonatan Bisk**[1], **Katerina Fragkiadaki**[1]
[1]Carnegie Mellon University, [2]National Taiwan University

## ABSTRACT

The pursuit of robot generalists, instructable agents capable of performing diverse tasks across diverse environments, demands rigorous and scalable evaluation. Yet real-world testing of robot policies remains fundamentally constrained: it is labor-intensive, slow, unsafe at scale, and difficult to reproduce. As policies expand in scope and complexity, these barriers only intensify, since defining "success" in robotics often hinges on nuanced human judgments of execution quality. We introduce RobotArena ∞, a new benchmarking framework that overcomes these challenges by shifting VLA evaluation into large-scale simulated environments augmented with online human feedback. Leveraging advances in vision-language models, 2D-to-3D generative modeling, and differentiable rendering, our approach automatically converts video demonstrations from widely used robot datasets into simulated counterparts. Within these digital twins, we assess VLA policies using both automated VLM-guided scoring and scalable human preference judgments collected from crowdworkers, transforming human involvement from tedious scene setup, resetting, and safety supervision into lightweight preference comparisons. To measure robustness, we systematically perturb simulated environments along multiple axes, including textures and object placements, stress-testing policy generalization under controlled variation. The result is a continuously evolving, reproducible, and scalable benchmark for real-world-trained robot manipulation policies, addressing a critical missing capability in today's robotics landscape. Benchmark website at `robotarenainf.github.io`.

## 1 INTRODUCTION

While recent years have witnessed substantial progress in developing more capable and general robot policies, their evaluation remains a persistent challenge and lacks standardization. This problem becomes especially acute as policies grow more generalist, requiring broader and more diverse evaluation scenarios. **Real-world evaluation is inherently unscalable**: it is limited by logistics, safety concerns, and reproducibility issues, and requires significant human involvement for setup, execution, and scoring. Human operators must supervise trials and manually reset scenes, which restricts the scale and frequency of evaluations (Vincent et al., 2024; Abou-Chakra et al., 2025; Li et al., 2024c). Such manual oversight also raises concerns about consistency and fairness, particularly when baselines and new models are compared under slightly different conditions.

Centralized physical evaluation provides a gold standard, where policies are tested under identical conditions, typically by submitting containers or shipping robots to a shared testing site. Notable examples include the Amazon Picking Challenge Correll et al. (2016), the Open-Vocabulary Mobile Manipulation Challenge Yenamandra et al. (2023), and RoboCup@Home Matamoros et al. (2019). However, the high cost for both organizers and participants means such events occur infrequently, often no more than once a year. In contrast, fields such as computer vision and natural language processing have advanced rapidly thanks to standardized benchmarks that provide consistent metrics, clear performance targets, and a foundation for fair comparison across methods Deng et al. (2009); Schuhmann et al. (2022). Our work is particularly inspired by LMarena Chiang et al. (2024), a large-scale, crowdsourced evaluation framework that benchmarks LLMs and VLMs through direct pairwise

---

*Equal contribution

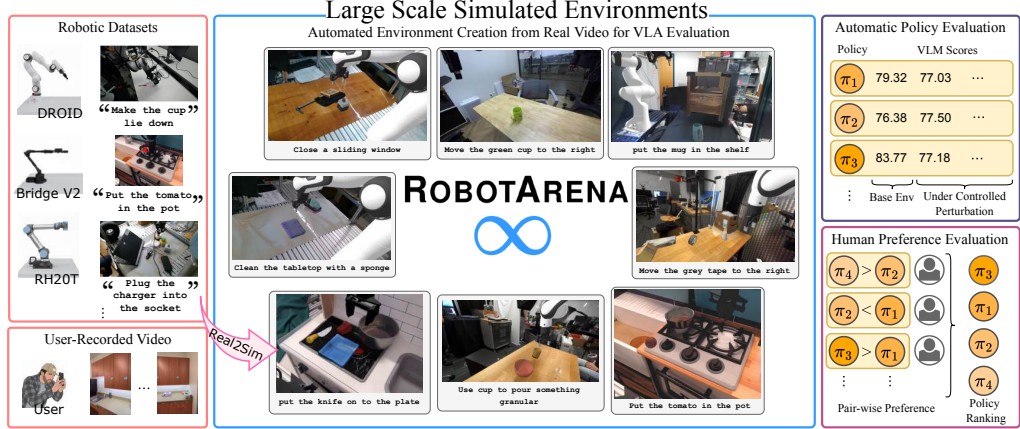

Figure 1: **RobotArena ∞ provides a scalable and extensible robot benchmarking framework by automating environment construction and evaluation.** It automatically generates simulated environment seeded from real videos, deploys robot policies, and evaluates them using VLMs and crowdsourced workers that cast preferences between pairs of execution videos. The simulated environments are derived from both in-distribution and out-of-distribution videos, enabling rigorous tests of generalization in contemporary robot policies.

comparisons of responses to the same prompt by human annotators. By aggregating thousands of such head-to-head matchups across diverse prompts, LMarena produces an Elo-style ranking that reflects collective judgments of model quality. Motivated by this success, we ask: **What would be the analogue of LMarena for Robotics?**

**RobotArena ∞:** We introduce RobotArena ∞, a new benchmarking framework that scales robot evaluation by deploying policies in automatically constructed simulated environments and assessing them through automatic VLM score and online human preference feedback. We first automatically translate real videos into corresponding simulation environments, building upon recent advances in vision-language models for scene understanding, 2D-to-3D generative models for asset creation, and differentiable rendering for camera pose estimation. We also introduce systematic environment perturbations, such as variations in lighting, object placement, and background appearance. We then deploy VLAs in these environments and evaluate their execution trajectories using two complementary strategies: (1) absolute evaluation, in which prompted VLMs or crowdsourced human workers estimate task progress scores for each video frame, and (2) relative evaluation, in which human annotators cast pairwise preferences to execution videos from different robot policies performing the same task. We measure both **in-distribution** performance by testing on simulation environments seeded from training videos in established datasets such as Bridge Walke et al. (2023), and **out-of-distribution** performance, by testing on environments generated from videos outside the training set. Our initial benchmark aggregates more than 8500 preference pairs across one hundred nominal environments and hundreds of perturbations, comparing six VLAs from independent labs worldwide. To the best of our knowledge, this constitutes the largest-scale robot evaluation effort to date.

Our robot evaluations reveal several key insights. First, vision–language–action (VLA) models are highly sensitive to dataset differences: performance drops when they are tested in environments outside their training distribution, indicating that current models are not true generalists but instead specialize to the environments represented in their training data, consistent with findings in Xing et al. (2025). Second, even within the same environment, performance degrades under perturbations, showing that robustness to distribution shifts remains an open challenge. At the same time, model rankings are consistent across conditions, suggesting that architectural and data design choices lead to measurable and reproducible performance differences.

RobotArena ∞ is inspired by prior efforts to design scalable robot benchmarks, particularly the seminal contributions of BEHAVIOR (Li et al., 2024) and SIMPLER (Li et al., 2024c). BEHAVIOR boasts an impressive manual effort of asset and environment creation, while SIMPLER reconstructs four real-world Bridge scenes and includes hand-designed reward functions. Compared to these ef-

forts, RobotArena ∞ offers a far more scalable and extensible framework by automating environment construction and evaluation.

In summary, our contributions are as follows:

1. We present **a scalable and extensible benchmarking protocol for robotics**, by coupling physics engines, real-to-sim translation and human preference feedback.
2. We introduce **a fully automated reality-to-simulation translation pipeline** built upon VLMs, 2D-to-3D generative models and differentiable rendering.
3. We evaluate **VLAs from labs worldwide across hundreds of environments with thousands of human preferences**, the most extensive robot evaluation to date.
4. We present **key evaluation results** that reveal how current robot policies generalize—or fail to—under distribution shifts.

Our benchmark is not without limitations. We outline these and discuss future directions and extensions. Importantly, RobotArena ∞ will continue to benefit from advances in physics engines and real-to-sim research. Both our benchmark environments and evaluation code will be publicly released and centrally maintained for continual support.

## 2 RELATED WORK

**Evaluating Robot Policies in The Real World**    Evaluating robot policies in the real world in a fair, comprehensive, and reproducible way remains a major challenge. Most methods are evaluated in custom lab-specific settings using proprietary hardware, task definitions, and success metrics, which makes cross-institution comparisons difficult. While standardized benchmarks exist for focused areas such as grasp prediction (Fang et al., 2020) and motion planning (Moll et al., 2014; Chamzas et al., 2022), extending these to cover generalist skills remains complex. Standardization efforts using shared object sets (e.g., YCB (Calli et al., 2015), IKEA (Heo et al., 2023), NIST (Kimble et al., 2020), ACRV (Leitner et al., 2016)) help reduce some variability, but differences in hardware, camera placement, lighting, and workspace setup still hinder consistent evaluation across labs. Real-world evaluation is often time-consuming and labor intensive. Human operators are typically required to supervise trials and manually reset scenes, limiting the scale and frequency of evaluations. For example, Chi et al. (2024) report manually aligning a T-shaped object into 20 predefined start configurations for each rollout, a process repeated across all baselines. This reset step is non-parallelizable and presents a significant bottleneck when evaluating policies across different tasks, agents, and environments (Vincent et al., 2024; Abou-Chakra et al., 2025). While recent systems such as AutoEval (Zhou et al., 2025) aim to automate evaluation, they are often limited in scope—e.g., supporting only five tasks in three static real-world scenes. RoboArena Atreya et al. (2025) builds a system of distribution real-world evaluation where human users reset the scene, run the robot policies and evaluate the resulting robot executions. It targets scenes from the DROID environment and evaluates a set of policies finetuned on DROID. As real-world robot datasets and generalist policies continue to expand in scale and complexity, the need for a more scalable, general, and continuously evolving evaluation framework becomes increasingly urgent.

**Evaluating Robot Policies in Simulation**    Simulation offers a scalable and safe alternative for evaluating robot policies, with numerous benchmarks like: RLBench James et al. (2020), Colosseum (Pumacay et al., 2024), CALVIN (Mees et al., 2022), LIBERO (Liu et al., 2024a), PerACT2 (Grotz et al., 2024), Meta-World (Yu et al., 2020), IKEA Simulation (Lee et al., 2019), and Behavior-1K (Li et al., 2024). However, these typically assume policies are trained and tested in the same simulated environments, potentially favoring specialist policies that exploit advantages of the closed-world environments over generalist models trained on real-world or more general simulation data. Our approach, in contrast, uses simulation strictly as an evaluation environment, independent of the policy's training origin. We contend that simulation is increasingly viable for policy evaluation due to improving physics engines and advancements in generative models and VLMs, which can automate scene asset creation and task success detection. RobotArena ∞ leverages these by automatically generating diverse simulation-based evaluation environments. The closest work, SIMPLER Li et al. (2024c), creates high-fidelity replicas of real scenes with human efforts, showing strong sim-to-real correlation. Unlike SIMPLER (Li et al., 2024c), RobotArena ∞ automates both scene generation

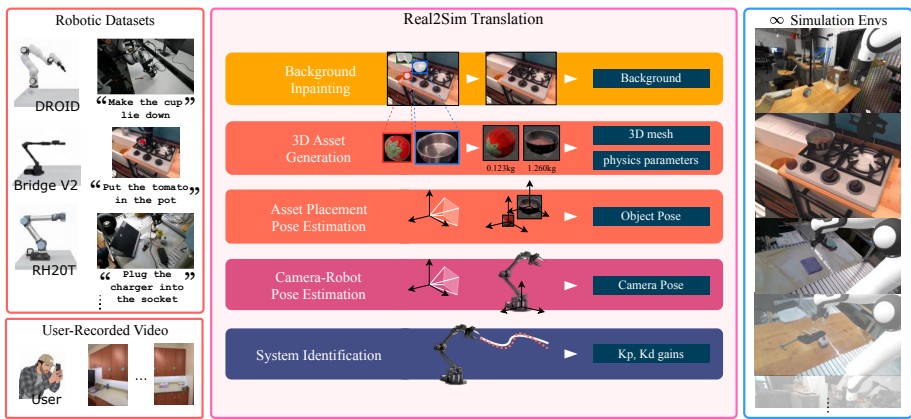

Figure 2: **Automated video-to-simulation translation** in RobotArena ∞. Given a frame from a video demonstration, we automatically create a corresponding simulated environment.

and task evaluation across many more tasks and environments, and evaluates policy robustness and generalization through systematic perturbations.

**Automated Real-to-Sim Translation** Recent efforts in real-to-sim translation, such as Phone2Proc Deitke et al. (2022), Scalable Real2Sim Pfaff et al. (2025), and Re3Sim Han et al. (2025), have demonstrated high-fidelity asset generation and physics-aware reconstruction. However, these approaches typically rely on multi-view captures, curated object libraries, or the use of fiducial markers, making them impractical for reconstructing typical single-view robot datasets. In contrast, RobotArena ∞ requires only a single RGB image from a static-camera demonstration. Our real-to-sim translation pipeline also differs from systems like RialTo Torne et al. (2024), which necessitates human-in-the-loop segmentation or specific calibration trajectories performed by the robot.

## 3 TRANSLATING VIDEOS TO SIMULATION FOR POLICY EVALUATION

RobotArena ∞ automatically creates simulation environments in physics engines from video demonstrations, as shown in Figure 2. Each robot demonstration is typically annotated with a language task description and per frame robot joint angle trajectories.

We also know the robot used and assume access to its URDF file. Our method extracts five key elements from the demonstration video: (1) the camera's 6-DoF pose relative to the robot body frame, (2) 3D mesh reconstructions of task-relevant objects, (3) their orientations, sizes, and material properties, (4) a clean background image, (5) proportional–derivative control gains. Together, these components enable realistic, physics-consistent simulations derived directly from video, without requiring manual calibration or curated annotations. The advantage of our modular design is long-term upgradability; each module can be replaced with stronger models as real-to-sim technologies improve, continually reducing error and improving benchmark fidelity.

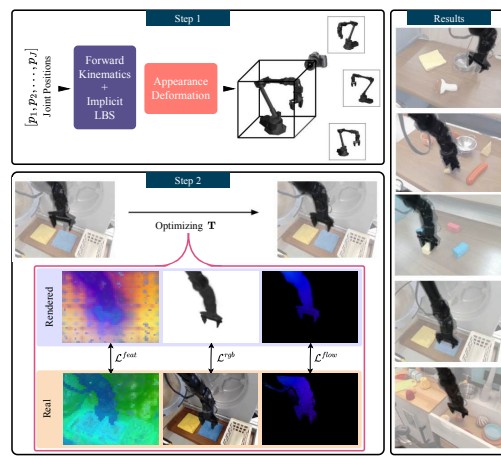

Figure 3: Automated robot-camera calibration through differentiable rendering of pose-conditioned 3D robot Gaussians.

**Automated Robot-Camera Calibration through Differentiable Robot Rendering** Robot demonstration videos are typically uncalibrated, that is, the pose of the camera with respect to the robot's frame is unknown. We estimate the camera-to-robot transformation using differentiable rendering, shown in Figure 3. Specifically, we first construct a joint angle–conditioned 3D Gaussian model of the robot via differentiable rendering in simulation based on its URDF file, following DR-Robot (Liu et al., 2024b), as shown in Figure 3 Step 1.

Given a robot demonstration video annotated with per-frame joint angles, we then render the Gaussian robot model and optimize the camera's 3D translation and orientation to minimize a composite alignment loss with three terms: (i) an RGB loss penalizing pixel-level appearance differences, (ii) a flow loss enforcing consistency between rendered motion fields and optical flow from the video Karaev et al. (2024), and (iii) a feature loss aligning DINOv2 embeddings between rendered and observed frames, shown in Figure 3 Step 2. When calibration metadata is available (e.g., from DROID (Khazatsky et al., 2024)), it is used for initialization; otherwise, we perform a coarse grid search to provide a robust starting point, as in BridgeV2 (Walke et al., 2023) and RoboMind (Wu et al., 2024). More details on robot-camera calibration can be found in Appendix B.

**Object and Scene 3D Reconstruction, Completion, and Physics Estimation** We prompt Gemini Team et al. (2023) to segment the robot and all task-relevant objects (Appendix C.1). Each segmented image crop is super-resolved with InvSR Yue et al. (2024) and converted into a textured 3D mesh using Hunyuan-3D (Team, 2025), which, like most image-to-3D mesh generation models, reconstructs objects in a canonical frame. To recover each object's correct 3D pose, we render 2D image views of the reconstructed 3D mesh and compare them against the 2D object crop using correspondence estimation from MINIMA (Ren et al., 2024). The view with the most feature matches is selected, and these correspondences are lifted to 3D using monocular depth estimate for the real image and simulated depth for the rendered view. Specifically, we first generate a relative depth map of the scene using MoGE (Wang et al., 2024). We then derive a metric scale factor by calibrating the robot arm's relative depth against its ground-truth depth from simulation, allowing us to unproject the masked pixels into an accurate metric-scale 3D point cloud. The final pose is then inferred via singular value decomposition (SVD) on the resulting 3D–3D correspondences (Appendix C.2). Physical and material properties for the objects are inferred by prompting Gemini and are then incorporated into the simulation environment.

To complete the scene, we generate a static background by inpainting the robot and object regions in the first video frame using the LaMa inpainting model Suvorov et al. (2021), producing a clean backdrop for the reconstructed assets (Appendix D). Finally, to accurately reproduce robot dynamics, we perform system identification to tune the proportional derivative (PD) controller gains, aligning simulated end-effector trajectories with those observed in the robot demonstration video (Appendix E). Apart from robot joint trajectory annotations obtained through teleoperation, our method requires no additional human supervision.

**Controllable Domain Perturbations** We introduce controlled perturbations to the generated environments in order to stress-test policy generalization under changes in background, scene arrangement, and color shifts. Specifically, we consider:

- **Background Change** ($\Delta$BG): Replaces the original scene background with different inpainted textures drawn from a diverse background dataset, isolating the policy's dependence on contextual appearance cues (Appendix G.1).
- **Color Shift** ($\Delta$Color): Alters the RGB channel configuration of the scene (e.g., converting RGB to BGR), applied at intensities from 0% to 100% in increments of approximately 33%, to test robustness against low-level color variation (Appendix G.2).
- **Object Pose Change** ($\Delta$ObjPose): Randomly permutes the location of objects in the scene (Appendix G.3).

## 4 Evaluating Robot Trajectories with Humans and VLMs

We evaluate robot execution videos automatically, by prompting VLMs (Section 4.1), and through crowdsourcing human preferences (Section 4.2).

## 4.1 EVALUATING ROBOT TASK PROGRESS SCORES WITH VLMs

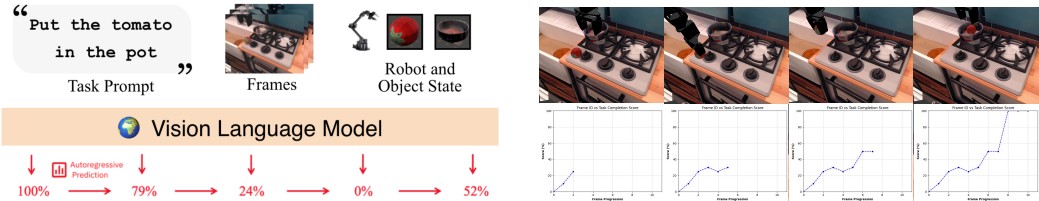

Figure 4: *Left*: Task progress scores computed by prompting Gemini 2.5 Pro with image frames and synchronized object and robot state sequences. *Right*: Example task progression. Top row shows execution frames; bottom row shows predicted progress scores.

We automate success detection and task progress evaluation at scale we prompting proprietary VLMs using video and privileged simulation state information.

Given a trajectory, the VLM is provided with the ordered sequence of video frames together with simulation state, including object states and robot states. The initial state is explicitly included as a zero-progress reference. The model is prompted to assign a progress score to each frame conditioned on both visual observations and state information (Figure 4). This multi-modal conditioning allows the evaluator to reason jointly about visual task completion, object configurations, and robot interactions.

To obtain a trajectory-level progress estimate, we compute the mean VLM score over the final 30% of frames in the trajectory. Intuitively, this focuses evaluation on the terminal phase of execution, where task completion (or failure) is most evident. We find that this metric aligns well with human-annotated progress scores and adopt it for all subsequent analysis.

## 4.2 EVALUATING ROBOT PERFORMANCE WITH HUMAN PREFERENCE FEEDBACK

While automated scoring provides scalability, human preference feedback is essential to capture nuanced aspects of robot behavior that numerical metrics may overlook. Our human evaluations are conducted through pairwise, double-blind comparisons of two policy execution videos obtained from the same simulation environment, under identical initial conditions and task instructions, following the protocol of Chiang et al. (2024). For each comparison, evaluators provide two forms of feedback: (i) a preference label specifying which policy performed better overall or whether they are tied, and (ii) a free-form natural language explanation describing the rationale behind their choice. We use free-form explanations as a way to increase evaluator engagement and attention. We found that requiring written justification led to more accurate human annotations. We have included the web interface for pairwise comparisons in the Appendix J and project page.

### 4.2.1 GLOBAL RANKING FROM PAIRWISE HUMAN PREFERENCES

We aim to compute a global policy ranking from pairwise human preferences. Let the set of $N$ policies be $\Pi = \{\pi_1, \dots, \pi_N\}$, and the dataset of pairwise comparisons be $\mathcal{D}_p = \{(P_{\pi_A, \pi_B}, t)\}$, where $P_{\pi_A, \pi_B} \in \{-1, 0, 1\}$ indicates a preference for $\pi_A$ over $\pi_B$, a tie, or a preference for $\pi_B$ over $\pi_A$, and $t$ denotes the task on which the comparison was made. Our goal is to derive a global ranking $\mathcal{R}$ over policies, e.g., $\pi_i \succ \pi_j \succ \dots \succ \pi_k$. While ties are recorded for completeness, we exclude them from the ranking objective and use only decisive comparisons ($\pm 1$) when fitting the model.

To aggregate pairwise judgments, we adopt the Bradley–Terry (BT) model Bradley & Terry (1952), a standard probabilistic framework for inferring rankings from paired comparisons. The model assigns each policy a latent ability score $\theta_i > 0$ and defines the probability that $\pi_i$ is preferred over $\pi_j$ as:

$$P(\pi_i \succ \pi_j) = \frac{\theta_i}{\theta_i + \theta_j}.$$

We estimate $\theta = \{\theta_1, \dots, \theta_N\}$ by maximizing the likelihood over all non-tied comparisons. This objective is concave in the log-parameterization, allowing efficient optimization via standard gradient ascent. The resulting ability scores yield a global ranking $\mathcal{R}$ by simple sorting. To quantify uncertainty

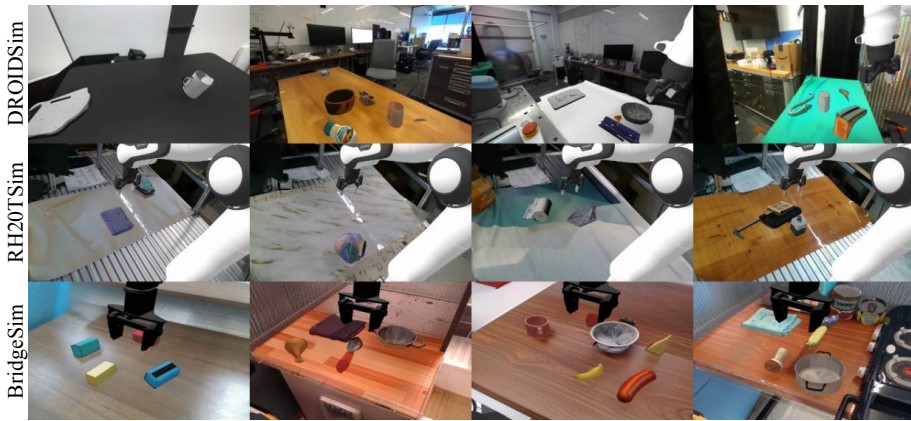

Figure 5: Simulation environments in RobotArena ∞ seeded from videos demonstrations in the datasets of Bridge, RH20T and DROID.

in the estimated rankings, we compute confidence intervals for $\theta$ using a robust (sandwich) variance estimator. Additional details on the ranking procedure and uncertainty estimation are provided in Section J.

## 5 BENCHMARKING ROBOT POLICIES IN ROBOTARENA ∞

**Generated Simulation Arenas** Our benchmark includes both in- and out-of-distribution environments and tasks for current VLAs, by selecting videos from within or outside their demonstration training sets, shown in Figure 5:

1. *BridgeSim* contains simulated environments and tasks generated from robot demonstrations in the BridgeV2 dataset Walke et al. (2023), a widely used subset of the Open X-Embodiment (OXE) dataset O'Neill et al. (2024), frequently employed to pretrain generalist robot policies.

2. *DROIDSim* consists of environments and tasks created from demonstrations in the DROID dataset Khazatsky et al. (2024). Unlike BridgeV2, DROID is often excluded from pretraining pipelines for generalist policies due to its higher noise levels Ma et al. (2024).

3. *RH20TSim* includes environments and tasks derived from the RH20T dataset Fang et al. (2023). Notably, among the candidate policies we evaluate, only *SpatialVLA*'s backbone has been pre-trained on this dataset.

**Candidate Policies to Evaluate** We benchmark the following open-source robot policies. All policies operate with a fixed egocentric camera and do not make use of wrist-mounted cameras[1]:

1. *Octo* Octo Model Team et al. (2023) is a transformer-based manipulation policy pre-trained on 800k demonstrations from the OXE dataset O'Neill et al. (2024). For our experiments, we evaluate the 93M-parameter Octo-Base model.

2. *RoboVLM* Li et al. (2024b) extends vision-language models (VLMs) into vision-language-action (VLA) policies by adding a continuous action prediction head. We evaluate the variant built on the KosMos backbone Peng et al. (2023).

3. *SpatialVLA* Qu et al. (2025) augments standard VLAs with 3D spatial reasoning through Ego3D Position Encodings and Adaptive Action Grids. It is trained on 1.1 million real-world robot demonstrations.

4. *CogAct* Li et al. (2024a) combines a 7B-parameter VLM backbone with a diffusion transformer for action prediction. It is pre-trained on the large-scale OXE dataset (22.5M frames, 60 datasets, 22 robot embodiments) and further fine-tuned on smaller real-world datasets for embodiment-specific adaptation.

---

[1]For $X$-$VLA$ and $\pi_0$ evaluation, we used the single view fine-tuned policy.

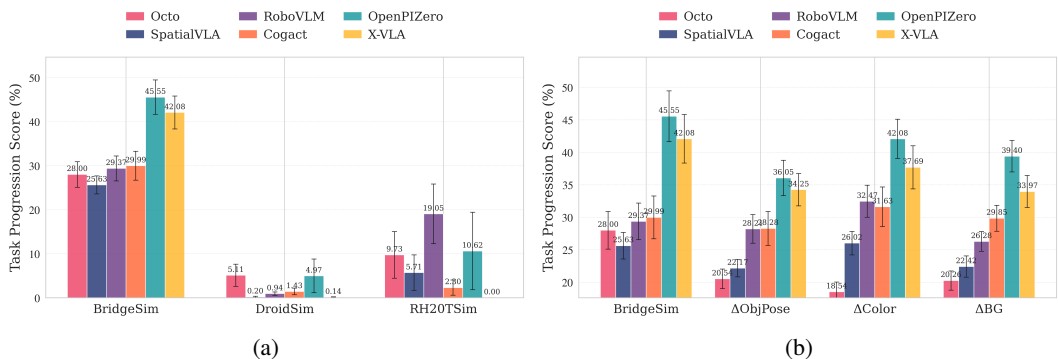

(a)                                                                 (b)

Figure 7: **Policy evaluation results** obtained from VLMs (a) in all RobotArena ∞ environments and (b) in perturbations of *BridgeSim* environments.

5. X-VLA Zheng et al. (2025) introduces a scalable cross-embodiment VLA framework built on a soft-prompted Transformer architecture. It uses embodiment-specific learnable soft prompts while sharing a unified backbone trained with a flow-matching objective. We evaluate the 0.9B-parameter X-VLA-0.9B model, pretrained on 290K heterogeneous episodes spanning seven hardware setups, and adapted using parameter-efficient finetuning.

6. $\pi_0$ Black et al. (2024) is a vision-language-action (VLA) foundation model built on a pre-trained vision-language backbone (PaliGemma) and augmented with a flow-matching action expert for continuous control. We evaluate the open-source reimplementation, `open-pi-zero` allenzren (2025), which reproduces the $\pi_0$ architecture and training recipe.

## 5.1 ROBOT BENCHMARKING RESULTS IN ROBOTARENA ∞

We show the VLA rankings derived from human pairwise preferences in Figure 6, along with confidence intervals computed using a robust sandwich variance estimator. We also show the automated VLM-derived evaluation results split across *BridgeSim*, *DROIDSim*, and *RH20TSim* in Figure 7 (a) with Standard Error of the Mean (SEM). We also show the performance in perturbed environments in Figure 7 (b) with SEM for the *BridgeSim* for which the policies perform the best. Human preferences align with the VLM task progress scores: $\pi_0$ and X-VLA are preferred more often than Cogact, RoboVLMs, Octo and SpatialVLA. The human and VLM rankings match exactly, indicating perfect agreement between automated and human evaluation.

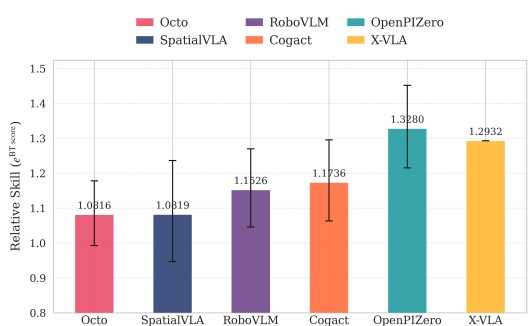

Figure 6: **Human preference ranking of VLAs in BridgeSim environments** from 8,749 pairwise comparisons.

We draw the following conclusions from the evaluations:

(1) **Cross-dataset generalization is weak:** Policies perform substantially worse in environments derived from datasets they were not trained on (e.g., DROID and RH20T), indicating current VLAs are not true generalist.

(2) **Model choice matters:** The benchmarking results in *BridgeSim* and its perturbed variants clearly distinguish $\pi_0$ and X-VLA as the top-performing architectures. However, their performance is not universally superior. For example, in RH20TSim, RoboVLM (19.05%) achieves a substantially higher score than all other models, while X-VLA fails (0.00%).

(3) **The "Spatial Paradox":** $\pi_0$ and X-VLA might have gained implicit 3D structure because their pre-training includes wrist-camera views. This suggests that the cross-view consistency learned from

raw multi-view data may provide a more robust spatial prior for manipulation than the current explicit 3D inductive biases used in SpatialVLA.

(4) **Backbone strength drives robustness:** Policies with stronger VLM backbones are more resilient to color perturbations. This suggests that these models rely more on invariant structural cues than on the superficial appearance features that distract less generalist models like Octo.

(5) **Overfitting to specific task configurations:** Performance across all evaluated policies degrades when backgrounds are perturbed or when object poses are randomized. This decline indicates that current VLAs may overfit to the specific visual and spatial setups of their training datasets rather than achieving true semantic generalization. While 3D structural modeling provides some resilience to pose changes, the sensitivity to background suggests that policies still rely heavily on fixed environmental cues.

## 5.2 POLICY RANKING IN ROBOTARENA ∞ VERSUS THE REAL WORLD

Estimation of correlations between policy rankings in RobotArena ∞ and in the real world would require multi-task evaluations of robot policies in real-world environments that recreate the precise structures, textures and camera arrangements from environments in RobotArena ∞ (that is, videos in Bridge, DROID or RH20T), a task very difficult to execute accurately, free of human biases. In fact, it is precisely real world policy evaluation that our framework attempts to replace. However, we did test such correlation for one task, *"Put the carrot in the plate"*, by recreating the same scenes in simulation and the real world and deploying RoboVLM, Octo, and SpatialVLA in both. RoboVLM and SpatialVLA succeeded in both real and simulated settings, while Octo failed in both, consistently attempting but failing to grasp the carrot, as shown in Figure 8.

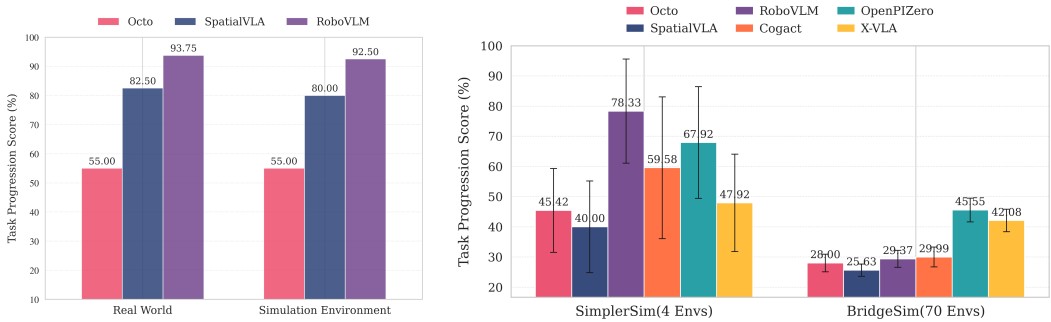

Figure 8: **Comparison of Simulation-Based Against Real-World-based Robot Evaluations** using Ma et al. (2024)

Figure 9: **Policy evaluation results in RobotArena ∞ versus SIMPLER of Li et al. (2024c).**

## 5.3 ROBOTARENA ∞ VERSUS SIMPLER OF LI ET AL. (2024C)

In Figure 9, we compare the performance of multiple VLAs on our reproduction of the four environments from the SIMPLER benchmark Li et al. (2024c), which are designed to approximate the conditions of the Bridge dataset, and on our BridgeSim benchmark, which contains 70 environments derived from the same Bridge V2 dataset. *All VLAs achieve substantially higher scores on SIMPLER than on BridgeSim.* This discrepancy suggests that evaluation on SIMPLER may overestimate policy performance due to the very limited number of test environments and the biased selection of scenarios. In contrast, BridgeSim's broader coverage and increased diversity make it a more rigorous and reliable benchmark for evaluating current and future generalist robot policies.

## 6 LIMITATIONS AND FUTURE DIRECTIONS

By leveraging recent advances in reality-to-simulation translation and crowdsourced evaluation, RobotArena ∞ provides a scalable and extensible robot benchmark. Our evaluators are not domain experts or robotics researchers, but everyday end-users—the very audience that robots are ultimately

intended to serve. The current benchmark has two main limitations. First, the policies evaluated do not yet incorporate wrist-camera inputs, which restricts the fidelity of certain manipulations. We are actively extending our reality-to-simulation pipeline to generate complete 3D interactive environments that will support multi-view observations. Second, current simulators still struggle to model fine-grained contact dynamics, such as inserting a charger into a socket. Despite progress in physics engines and automated asset generation Wang et al. (2025); Narang et al. (2022), these tasks remain difficult to reproduce faithfully, highlighting a crucial direction for future research. Looking forward, we believe RobotArena $\infty$ is well positioned to benefit from advances in simulation, physics engines, and environment generation, and to serve as a continually improving platform for evaluating the next generation of robotic foundation models.

## 7    CONCLUSION

We introduced RobotArena $\infty$, a new benchmarking framework that scales robot evaluation through automated reality-to-simulation translation and online human preference feedback. By coupling advances in vision–language models, 2D-to-3D generative models, and differentiable rendering, our framework enables the automatic construction of simulated environments directly from real-world videos. We demonstrated its utility by evaluating multiple VLAs across hundreds of environments and over 8500 human preference judgments, yielding the most extensive study of policy robustness and generalization to date. Our findings highlight significant lack of generalization across benchmarks, sensitivity to perturbations, and systematic differences across architectures—revealing both the limitations of current VLAs and the promise of scalable evaluation protocols. Looking ahead, we envision RobotArena $\infty$ as a continually evolving benchmark that grows alongside advances in robot learning. Future directions include expanding the range of tasks, incorporating more diverse real-world data sources, and leveraging improvements in physics engines and real-to-sim translation. By releasing both the benchmark environments and evaluation code, we aim to provide the community with an open, extensible platform for rigorous policy evaluation.

## ACKNOWLEDGEMENTS

We would like to acknowledge Pang-Chi (Sean) Lo for his continuous support in integrating new VLA models into the benchmark and improving the environments. Due to ICLR's policy on adding new authors after submission, we were unable to include him as a co-author. We would like to thank He Zhu for providing the VLM prompt that we used in our VLM-derived policy evaluations. We would also like to thank Ayush Jain for general discussion regarding scene segmentation in our real-to-sim pipeline. This work is funded by an Amazon AGI gift, SAFRON DARPA award HR0011-25-3-0203, an NSF Career award, and AFOSR Grant FA9550-23-1-0257.

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

# A  WEBSITE AND CODE

The project website is available at https://robotarenainf.github.io, and the source code can be accessed at https://github.com/offjangir/RobotArena.

# B  INFERRING CAMERA-ROBOT POSE TRANSFORMATION VIA ANALYSIS-BY-SYNTHESIS

In this optimization problem, we work with a sequence of video frames capturing the robot's motion, denoted $I_t^{gt}$, where $t$ ranges from 1 to $T$. We also have the robot's joint angles $q^t$ at each time $t$, provided by the dataset. Our task is to estimate the transformation $\mathbf{T} \in SE(3)$, which aligns the robot's coordinate system with the camera's. Using $\mathbf{T}$ and $q^t$, we produce rendered a image $I_t$, a synthetic depiction of the robot at time $t$. Motion is captured through optical flow $F_t^{gt}$, estimated between consecutive video frames $I_t^{gt}$ and $I_{t+1}^{gt}$, and rendered flow $F_t$. Additionally, we use the DINOv2 model to extract feature maps $\phi_t^{gt} = \text{DINOv2}(I_t^{gt})$ from the video frames and $\phi_t = \text{DINOv2}(I_t)$ from the rendered images, enhancing alignment with high-level image features.

The optimization aligns the rendered and observed data using three loss terms:

1. **RGB Loss**: Measures the color difference between the rendered and video images at each time $t$:
$$\mathcal{L}_{rgb}(t) = \|I_t - I_t^{gt}\|_2^2 \tag{1}$$

2. **Flow Loss**: Compares the rendered flow to the optical flow for each pair of frames from $t = 1$ to $T - 1$:
$$\mathcal{L}_{flow}(t) = \|F_t - F_t^{gt}\|_2^2 \tag{2}$$

3. **Feature Loss**: Aligns the feature maps using cosine loss at each time $t$:
$$\mathcal{L}_{feat}(t) = 1 - \cos\left(\phi_t, \phi_t^{gt}\right) \tag{3}$$

The total loss combines these terms over the sequence:
$$\mathcal{L} = \sum_{t=1}^{T} \lambda_{rgb}\mathcal{L}_{rgb}(t) + \sum_{t=1}^{T} \lambda_{feat}\mathcal{L}_{feat}(t) + \sum_{t=1}^{T-1} \lambda_{flow}\mathcal{L}_{flow}(t) \tag{4}$$

where $\lambda_{rgb}$, $\lambda_{feat}$, and $\lambda_{flow}$ are weights balancing each term. The optimal $\mathbf{T}$ is found by minimizing:
$$\mathbf{T}^* = \underset{\mathbf{T} \in SE(3)}{\operatorname{argmin}} \mathcal{L} \tag{5}$$

# C  RELEVANT OBJECT SEGMENTATION, 3D RECONSTRUCTION, AND MATERIAL PROPERTY ESTIMATION

The creation of simulation-ready 3D assets for movable objects is accomplished through a multistage pipeline. This process begins with semantic understanding of the scene using a vision language model (VLM), specifically Gemini Team et al. (2023). We prompt Gemini to segment the robot and all foreground objects it interacts with. The resulting segmentation masks provide fine-grained object localization. To enhance visual details critical for reconstruction, the original image is first super-resolved, then resized to its original dimensions. The segmentation masks are applied to isolate and crop each object of interest. These object-specific image patches are further refined using a second stage of super-resolution with InvSR Yue et al. (2024), producing high-fidelity inputs for 3D reconstruction. The enhanced patches are then processed by an image-to-3D model, Hunyuan-3D Team (2025), which outputs detailed meshes capturing both the geometry and texture of each object. To ensure physical plausibility within simulation, Gemini Team et al. (2023) is prompted to infer object-specific physical parameters such as mass and friction coefficients. Final asset preparation includes appropriate scaling and placement within the simulated environment to ensure consistent alignment with the original scene configuration.

This section describes the method of recovering the 3D position and orientation of objects from a single RGB image. The approach leverages camera parameters recovered by our previously described method, detailed in Section B, and an initial depth map estimated using (MoGE Wang et al. (2024)).

## C.1    OBJECT SEGMENTATION

To obtain accurate localization of task-relevant entities, we employ Gemini Team et al. (2023) with three structured prompts tailored for different goals.

**Robot Segmentation Prompt**    We first segment the robot, as jointly detecting all entities may result in missed or inaccurate detection of the robot. The prompt below is used to obtain the robot's segmentation mask:

> Give segmentation masks for the robot in the scene. Output a JSON list of segmentation masks where each entry contains the 2D bounding box in "box_2d", a descriptive text label in "label", and the mask in "mask".

**Foreground Object Segmentation Prompt**    After isolating the robot, we proceed to segment the foreground objects it interacts with, along with estimating their physical properties. The following prompt is used to obtain segmentation masks and associated physical attributes:

> Give segmentation masks for all important complete foreground objects on the plane which the robot interacts with. You must ignore any background objects, irrelevant surfaces, or any objects that are occluded, covered, or severely blocked by other objects.
> Output a JSON list of segmentation masks where each entry contains:
> - "box_2d": the 2D bounding box
> - "mask": the segmentation mask (in image format)
> - "label": a descriptive text label
> - "mass": estimated object mass in kilograms (float)
> - "friction": estimated friction coefficient (float)
> - "surface_type": one of [Glass, Water, Emission, Plastic, Rough, Smooth, Reflective, Metal, Iron, Aluminium, Copper, Gold]

**Task-Relevant Object Selection Prompt (conditional)**    If a task description is provided, we use the following prompt to analyze the previously segmented objects, identifying those with semantic relevance to the task and determining their functional roles (e.g., as a manipulation *target* or *destination*):

> Consider this specific task: {task}, given the object names: {object_names}, please select the object names that are relevant to the task and identify their role in the task from "target" and "destination".
> Output in this format exactly as: target: <target_object>, destination: <destination_object>.
> Please try to identify the target object (If you can't find an exact match, use the closest one from given object names) and if there is no destination object, only output the target object.

For example, given the task: *put the pot on top of the yellow cloth* and the segmented object names: *sauce pan, cloth*, the output would be: *target: sauce pan, destination: cloth*

## C.2    OBJECT POSE ESTIMATION

**Initial 3D Point Cloud Reconstruction**    With the 2D mask of the target object generated using Gemini Team et al. (2023), we unproject each masked pixel into a 3D point cloud in world coordinates. This process uses the recovered camera intrinsic and extrinsic parameters $\mathbf{K}$, $\mathbf{R}$, $\mathbf{t}$ and a metric-scale depth map $D(u, v)$. To recover metric depth maps, we compute the scale factor by comparing MoGE's Wang et al. (2024) relative depth estimates of the robot arm against the simulated robot arm's ground-truth depth; applying this factor converts the relative predictions into accurately scaled metric depth maps.

For each pixel $(u, v)$ within the mask, its world-space coordinate $\mathbf{P}_{\text{world}}$ is then given by

$$\mathbf{P}_{\text{world}} = \mathbf{R}^\top D(u, v) \, \mathbf{K}^{-1} \, [u, v, 1]^\top \, - \, \mathbf{t}.$$

**Simulated Viewpoint Generation and 2D Correspondence Establishment**    To robustly estimate the object's pose, we generate multiple synthetic views. This process begins with the generated 3D mesh of the object. To ensure dimensional consistency for the simulation, the scale of this generated mesh is aligned by utilizing its bounding box and the bounding box of the original point cloud. The scale-aligned mesh is then imported into our simulation environment and initialized at a reference position. In this simulated environment, multiple views are rendered by a camera programmed to navigate around this central object position. The camera's trajectory is systematically defined by varying its elevation and azimuth angles relative to the reference position.

Let

$$z_{\text{levels}} = \{z_1, z_2, \ldots, z_L\}, \quad \theta_{\text{values}} = \{\theta_1, \theta_2, \ldots, \theta_M\}, \quad N_\theta = M,$$

For each view index $i \in \{0, \ldots, N - 1\}$, we compute:

$$\theta_i = \theta_{\text{values}}\big[i \bmod N_\theta\big], \tag{6}$$

$$z_i = z_{\text{levels}}\big[\lfloor i/N_\theta \rfloor\big], \tag{7}$$

$$\hat{\mathbf{v}}_i = \frac{1}{\sqrt{\cos^2\theta_i + \sin^2\theta_i + z_i^2}} \begin{bmatrix} \cos\theta_i \\ \sin\theta_i \\ z_i \end{bmatrix}, \tag{8}$$

$$\mathbf{p}_i = \mathbf{p}_{\text{ref}} + d\,\hat{\mathbf{v}}_i. \tag{9}$$

where $\mathbf{p}_{\text{ref}}$ is the target's centroid in world coordinates and $d > 0$ is the fixed radial distance. The simulated camera is then oriented such that its viewing direction is aimed at $\mathbf{p}_{\text{ref}}$.

For each camera pose $\mathbf{p}_i$, we render a synthetic image:

$$I_{\text{sim}}^i = \text{Render}(\mathbf{p}_i) \, ,$$

We then apply the MINIMA algorithm Ren et al. (2024) to establish 2D keypoint correspondences between $I_{\text{sim}}^i$ and the original masked image. This produces, for each view $i$, a set of $M_i$ matched keypoint pairs:

$$\mathcal{C}_i = \big\{ \big(\mathbf{k}_{\text{sim}}^{i,j}, \, \mathbf{k}_{\text{orig}}^{i,j}\big) \big\}_{j=1}^{M_i}.$$

where $\mathbf{k}_{\text{sim}}^{i,j}$ is the $j$-th keypoint in $I_{\text{sim}}^i$ and $\mathbf{k}_{\text{orig}}^{i,j}$ is its corresponding keypoint in the original image.

**Optimal View Selection and 3D Keypoint Pair Generation**    We select the optimal view index

$$i^* = \underset{i \in \{0, \ldots, N-1\}}{\arg\max} \; |\mathcal{C}_i|,$$

and denote the corresponding rendered image by $I_{\text{sim}}^*$. The set of matched keypoints for this view is

$$\mathcal{C}_{i^*} = \big\{ \big(\mathbf{k}_{\text{sim}}^{*,j}, \, \mathbf{k}_{\text{orig}}^{*,j}\big) \big\}_{j=1}^{M},$$

The optimal 2D keypoint correspondences $\{(\mathbf{k}_{\text{sim}}^{*,j}, \mathbf{k}_{\text{orig}}^{*,j})\}_{j=1}^{M}$ are then lifted into 3D by unprojection.

Specifically, for each simulated keypoint $\mathbf{k}_{\text{sim}}^{*,j}$ we compute its 3D coordinate via

$$\mathbf{P}_{\text{sim},j} = \text{Unproject}\big(\mathbf{k}_{\text{sim}}^{*,j}; \, \mathbf{K}_{\text{sim}}^*, \, \mathbf{R}_{\text{sim}}^*, \, \mathbf{t}_{\text{sim}}^*, \, D_{\text{sim}}^j\big), \tag{10}$$

where $\mathbf{K}_{\text{sim}}^*, \mathbf{R}_{\text{sim}}^*, \mathbf{t}_{\text{sim}}^*$ are the intrinsic and extrinsic parameters of the known simulated camera and $D_{\text{sim}}^j$ is the depth rendered in the simulation.

Similarly, for each original image keypoint $\mathbf{k}_{\text{orig}}^{*,j}$ we obtain

$$\mathbf{P}_{\text{orig},j} = \text{Unproject}\big(\mathbf{k}_{\text{orig}}^{*,j}; \, \mathbf{K}, \, \mathbf{R}, \, \mathbf{t}, \, D(u_{\text{orig}}^j, v_{\text{orig}}^j)\big), \tag{11}$$

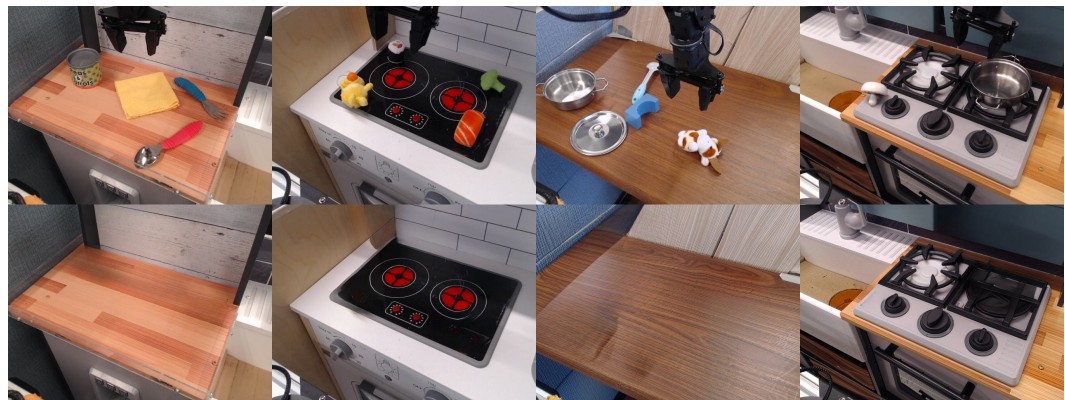

Figure 10: Background Inpainting Results. The top row shows the original RGB images with task-relevant objects present; the bottom row shows the corresponding inpainted images after foreground object removal, where only backgrounds remain.

where $\mathbf{K}$, $\mathbf{R}$ and $\mathbf{t}$ are the recovered camera intrinsic and extrinsic parameters, and $D(u_{\text{orig}}^{j}, v_{\text{orig}}^{j})$ is the aligned MoGE Wang et al. (2024) depth at pixel $(u_{\text{orig}}^{j}, v_{\text{orig}}^{j})$.

This yields a set of $M$ 3D–3D correspondences

$$\mathcal{S} = \left\{ (\mathbf{P}_{\text{sim},j}, \mathbf{P}_{\text{orig},j}) \right\}_{j=1}^{M}.$$

**Rigid Transformation Estimation using SVD**  Given the set of $M$ 3D–3D keypoint correspondences $\{(\mathbf{P}_{\text{sim},j}, \mathbf{P}_{\text{orig},j})\}_{j=1}^{M}$, our goal is to estimate the rigid transformation $(\mathbf{R}, \mathbf{t})$ that best aligns these two point sets. This transformation consists of a rotation matrix $\mathbf{R} \in \mathrm{SO}(3)$ and a translation vector $\mathbf{t} \in \mathbb{R}^3$, such that:

$$\mathbf{P}_{\text{orig},j} \approx \mathbf{R}\mathbf{P}_{\text{sim},j} + \mathbf{t}.$$

We employ the Singular Value Decomposition (SVD) method for this estimation, which yields the rotation $\mathbf{R}$ and translation $\mathbf{t}$ that map the 3D keypoints of the simulated object to the original point cloud and thus recover the position and orientation of the generated object.

## D  BACKGROUND INPAINTING

We generate a static background by inpainting the robot and object regions in the first video frame with LaMa Suvorov et al. (2021). We visualize the results of background inpainting for 4 different scenes in Bridge Dataset Walke et al. (2023) in Figure 10.

## E  SYSTEM IDENTIFICATION

To improve the simulation fidelity of the robot's open-loop motion, we apply system identification (SysID) to tune the proportional–derivative (PD) controller gains, $\mathbf{kp}$ and $\mathbf{kd}$. The goal is to minimize the discrepancy between simulated and real end-effector trajectories, using only the joint position commands as input. For each trajectory in a dataset, we extract the end-effector pose sequence and run a physics-based simulation using the same joint commands. The simulated robot is controlled with a PD controller, and the gains are optimized to minimize the difference in end-effector positions over time.

$$\mathrm{kp}^*, \mathrm{kd}^* = \underset{\mathrm{kp,kd}}{\operatorname{argmin}} \sum_{t=1}^{T} \left( \|\mathbf{x}_t^{gt} - \mathbf{x}_t\|_2 + \arcsin\left( \frac{1}{2\sqrt{2}} \|\mathbf{R}_t^{gt} - \mathbf{R}_t\|_F \right) \right) \tag{12}$$

where $\mathbf{x}_t^{gt}$ and $\mathbf{R}_t^{gt}$ denote the ground-truth end-effector position and orientation from the real world dataset, $\mathbf{x}_t$ and $\mathbf{R}_t$ denote those of the simulated robot, and $\| \cdot \|_F$ denotes the Frobenius norm.

Note that we exclude the gripper from the system identification process, as its state in the dataset is typically represented by a binary open/close signal, which does not support continuous control tuning.

To search for optimal parameters, we use a gradient-free Simulated Annealing (SA) strategy. In each iteration, a candidate set of gains is evaluated in parallel using the Genesis simulator Authors (2024). All environments are initialized simultaneously, and the same candidate gains are applied to each. At each step:

- Inverse kinematics is solved in batch to produce joint targets for each trajectory.

- PD control is applied in each environment using the current candidate ($\mathbf{kp}$, $\mathbf{kd}$).

- The mean Euclidean distance between the simulated and dataset end-effector positions is computed.

- The SA policy perturbs the parameters and accepts new candidates based on the change in average error.

Gains are optimized over 5000 steps of SA. Each iteration involves simulating all trajectories for a fixed number 5000 of steps (typically ), making the process feasible in parallel with GPU acceleration. We use the following parameter ranges for the search: $\mathbf{kp} \in [2000, 15000]$, $\mathbf{kd} \in [10, 2000]$.

A visual comparison for Bridge Dataset Walke et al. (2023) before and after system identification is shown in Figure 11, where the dataset (red) and simulated (blue) end-effector trajectories are overlaid on the XY plane.

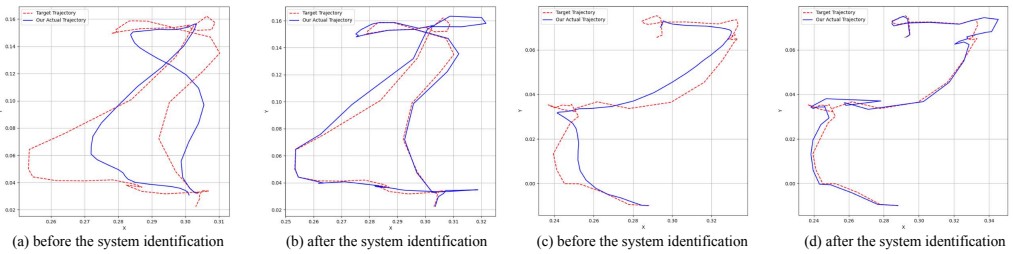

(a) before the system identification    (b) after the system identification    (c) before the system identification    (d) after the system identification

Figure 11: **Inferring robot control gains for matching robot trajectories in reality and simulation.** The end-effector trajectory, projected onto the XY plane, is shown for both the pre-identification (a,c) and post-identification (b,d) stages. After system identification, the simulated trajectory (blue) aligns closely with the recorded dataset trajectory (red), whereas significant deviations are observed prior to identification.

## F  *BridgeSim* ENVIRONMENT VISUALIZATIONS

We present additional visualizations of results from our Real2Sim pipeline in Figure 12.

## G  ENVIRONMENT PERTURBATIONS

To quantify policy robustness under realistic visual and spatial variability, we systematically introduce four controlled environment perturbations.

### G.1  BACKGROUND CHANGE (ΔBG)

For each scene, we replace the original background with five different inpainted backgrounds generated by our Reality-to-Simulation pipeline and keep all foreground objects exactly the same, so that for every scene we have five 'background flipped' variants that differ only in their contextual appearance, as illustrated in Figure 14.

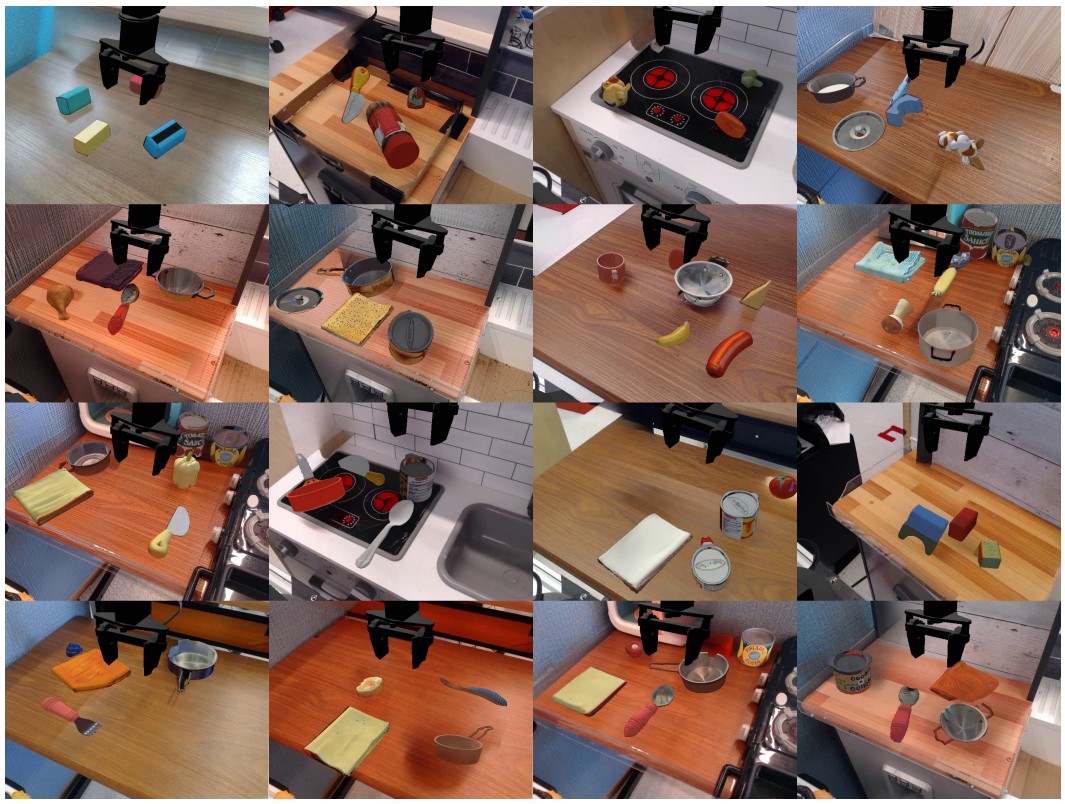

Figure 12: Visualizations of a subset of *BridgeSim* Environments

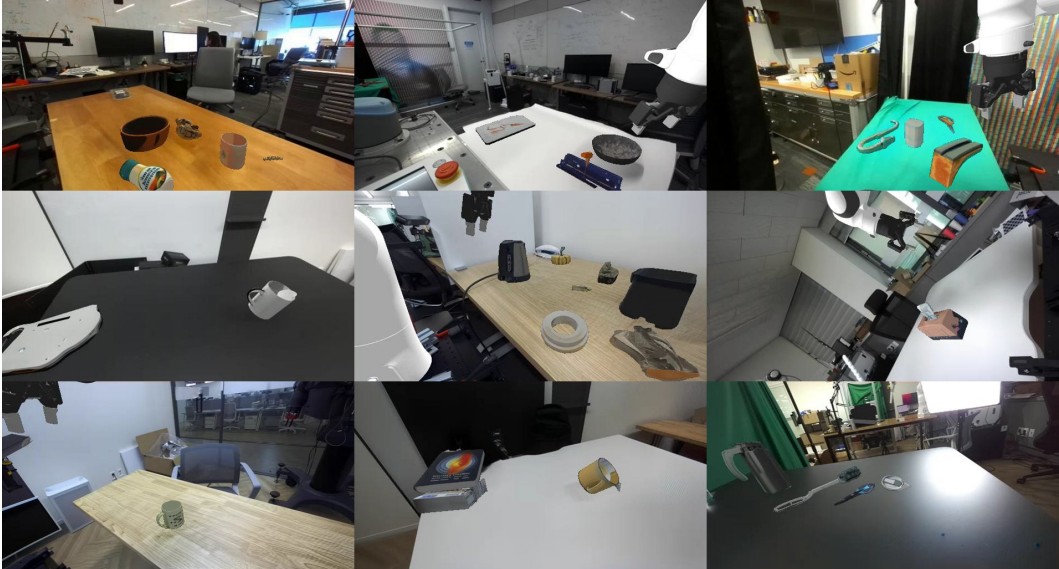

Figure 13: Visualizations of a subset of *DROIDSim* Environments

## G.2 COLOR SHIFT (ΔCOLOR)

For each scene, we leave the foreground (objects, lighting, dynamics) completely unchanged and apply only a low-level color perturbation to the background by remapping its RGB channels to BGR and blending back at four intensity levels. Concretely, for every background pixel with the original

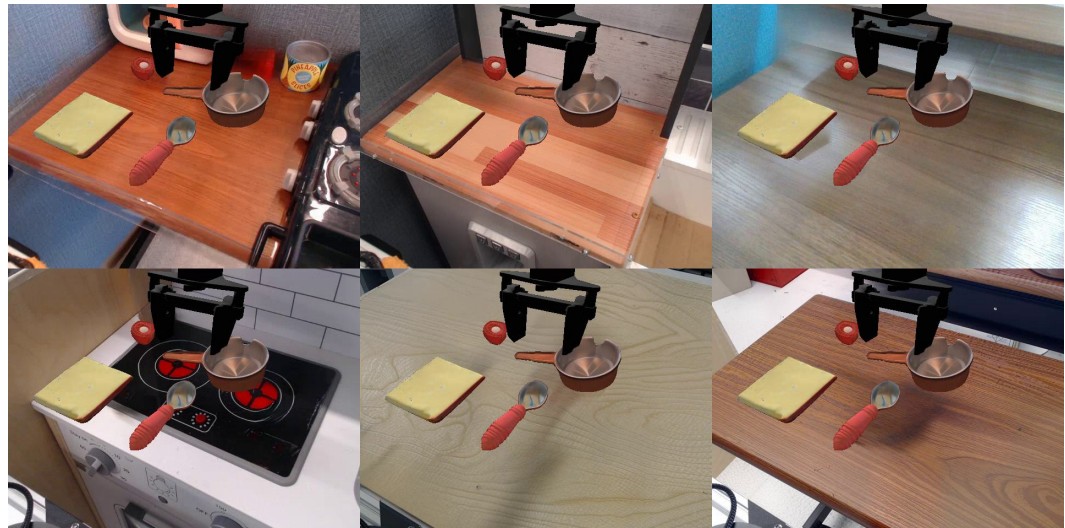

Figure 14: Background Change Example. The top-left image shows the original image without background perturbations.

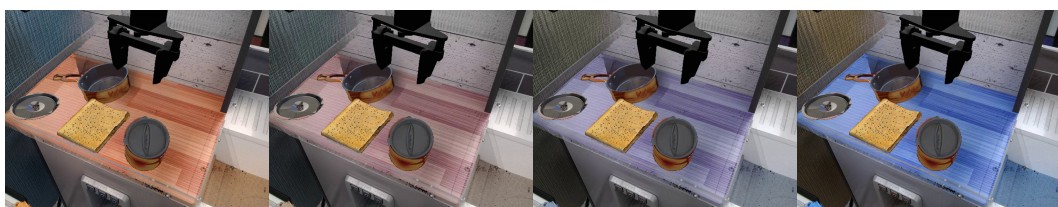

Figure 15: Color Shift Example. The leftmost image shows the original image without color perturbations.

color vector $[R, G, B]$, we compute:

$$C'(\alpha) \;=\; (1 - \alpha) \begin{bmatrix} R \\ G \\ B \end{bmatrix} \;+\; \alpha \begin{bmatrix} B \\ G \\ R \end{bmatrix},$$

where

$$\alpha \in \{0.00,\ 0.33,\ 0.66,\ 1.00\}$$

corresponds respectively to 0%, 33%, 66% and 100% BGR swap intensity. This yields four "color-swapped" background variants per scene, as illustrated in Figure 15.

### G.3 OBJECT POSE CHANGE ($\Delta$OBJPOSE)

For each scene containing $N$ distinct assets with original world-space positions $\{x_i\}_{i=1}^N$, we perform $N$ independent random permutations $\{\pi^{(k)}\}_{k=1}^N$ (including the identity permutation for the original layout). For permutation $k$, we reassign asset $i$ to the position of asset $\pi^{(k)}(i)$, yielding

$$x_i'^{(k)} = x_{\pi^{(k)}(i)}, \quad i = 1, \ldots, N.$$

This procedure generates $N$ scene variants that differ only in object arrangement, as illustrated in Figure 16.

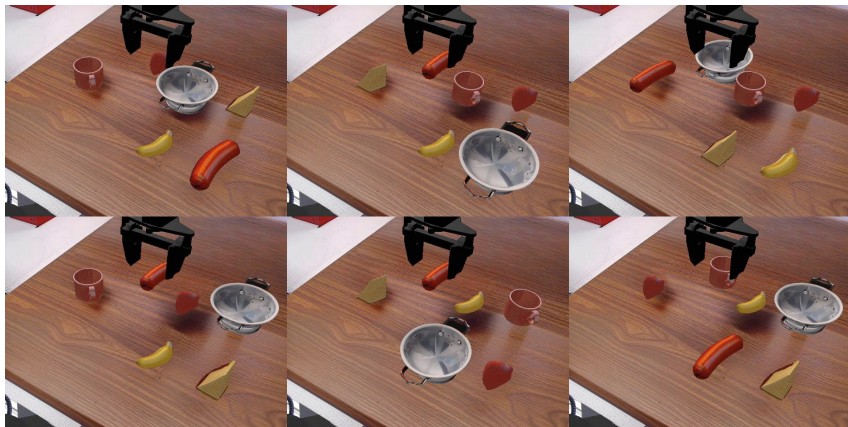

Figure 16: Object Position Perturbation Example. The top-left image shows the original setup without perturbation.

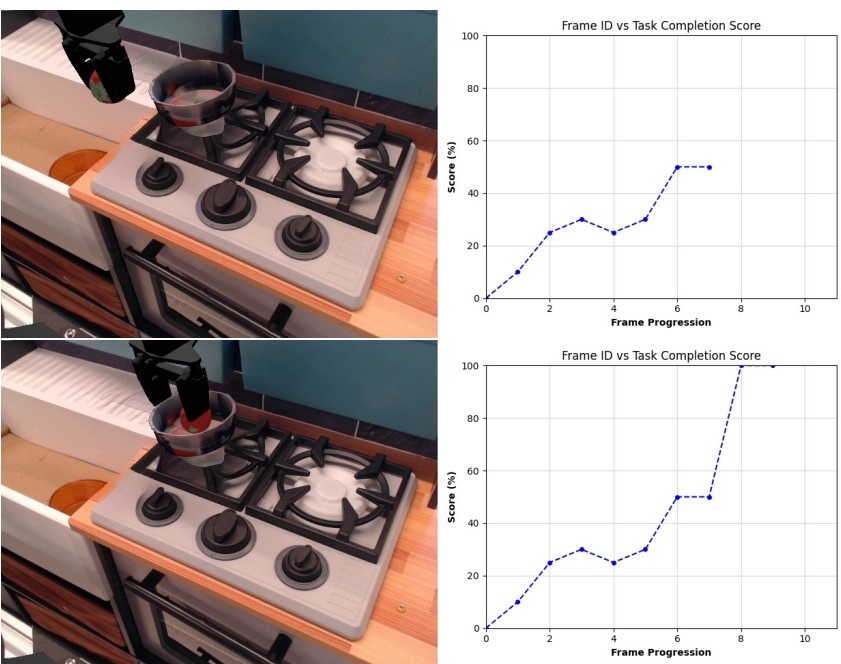

Figure 17: Example VLM-generated task evaluation curves on base environment. **Left panels:** Representative frames sampled at low- and high-progress points. **Right panels:** VLM-assigned completion score (y-axis) across frame ID (x-axis).

## H AUTOMATED TASK PROGRESS SCORING WITH VLMS

To obtain dense, fine-grained task progress signals, we use a structured evaluation prompt with a large Vision–Language Model (VLM), specifically Gemini. The prompt provides: (i) the natural language task instruction, (ii) structured robot and object state information, and (iii) a sequence of visual frames capturing the scene evolution. Concretely, we present both the initial and current scene states—each decomposed into visual observations, and action trajectory history. This multimodal formulation enables the VLM to jointly reason over visual evidence and structured state transitions when estimating task progress. The full prompt is provided below.

```
PROGRESS_EVAL_PROMPT = """You are an expert roboticist.  The robot is
performing the task of '{instruction}'.
Your task is to evaluate the robot's progress towards completing the
overall task by analyzing the provided initial/current scene states and
robot action trajectory.
***** Input *****
This is the initial state of the scene.
(1) Visual Observations:
{init_visual_observations}
(2) Object Layout:
{init_object_information}
(3) Contact Information:
{init_contact_information}
From the initial state, the robot has made progress represented by the
following trajectory:

{history_information}
Such progress results in the current state of the scene.
(1) Visual Observations:
{current_visual_observations}
(2) Object Layout:
{current_object_information}
(3) Contact Information:
{current_contact_information}
***** Output *****
Based on the above information, please do the following:
## Reflection
Evaluate the overall progress of the robot toward completing the task
by analyzing the trajectory and comparing initial and current scene
state.  Are the subgoals reasonable?  Is the robot making meaningful
progress towards the final goal?
## Progress Score
Predict a task completion percentage between 0 and 100.
- In the initial state, the task completion percentage is 0.  - If all
necessary steps are completed, the task completion percentage is 100.
- The score should increase when a meaningful subgoal is completed.  -
The score should decrease if an action undoes or degrades a previously
completed meaningful subgoal.  - The score should also decrease if the
robot's action causes SEVERE damage to other objects in the environment
(minor disturbance is acceptable and can be ignored).
In this part, just provide one score and one concise explaining
sentence.
Make sure you use '## Reflection' and '## Progress Score' as section
headers in your response.  Please use EXACTLY ONE number between 0 and
100 for the Progress Score.  """
```

We present the results for generative value learning based progressive score prediction in Figure 17
18 19.

## I    VLM EVALUATIONS

In the *RH20TSim* environment, Octo achieves a significantly higher VLM score than the other models.
Qualitative analysis reveals this is due to a minimalistic motion pattern; Octo just slowly lowers its
arm directly onto the workbench. This simple trajectory succeeds by coincidence, as the target object
happens to be positioned in the arm's path. This shortcut solution scores higher than the baseline
policies, which engage in more extensive but ultimately unsuccessful exploration.

## J    HUMAN EVALUATIONS

To assess the reliability of our automated task success metrics, we conducted a validation study by
comparing them against human judgments collected through a user study. We developed a web-based

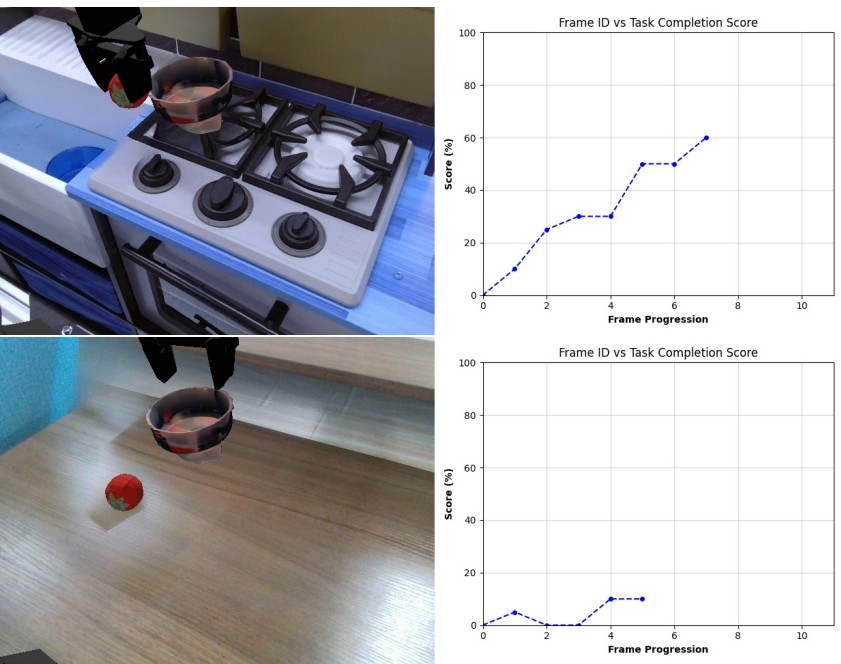

Figure 18: Example VLM-generated task evaluation curves on perturbed environments. **Top:** A high-progress moment immediately after the object lift, for which the VLM predicts a completion score of approximately 60%. **Bottom:** An inconsequential action with no strong effect on task progress, for which the VLM predicts a low completion score of approximately 10%.

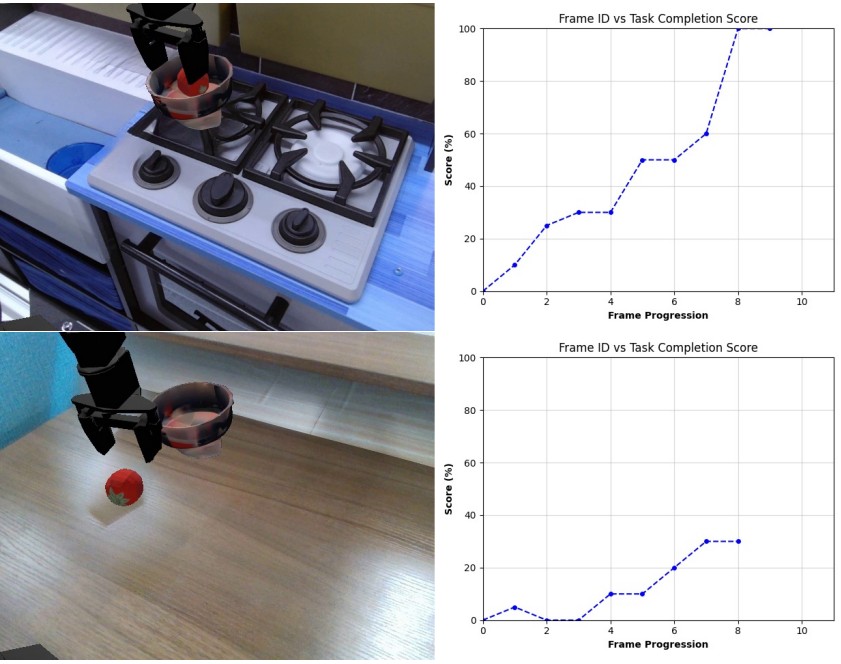

Figure 19: Example VLM-generated task evaluation curves on perturbed environments. **Top:** A successful pick-and-place execution—after the object lift the VLM score climbs steadily and correctly shows task completion. **Bottom:** An unsuccessful trajectory with completion score remaining low, demonstrating the VLM's capacity to detect failure to complete the task.

interface that presented annotators with two side-by-side videos, each showing a different policy attempting the same task described by a natural language instruction. For each pair, participants were first asked to provide descriptions of each robot's attempt and then judge which policy performed better or if it was a tie. An example of this interface is shown in Figure 20. To ensure high data quality, participants had to pass a qualification quiz by correctly evaluating at least 8 out of 10 video pairs. These pairs were intentionally chosen with clear performance differences, making them straightforward to judge.

From a total pool of 70 environments, we collected 8749 pairwise comparisons on the Amazon Mechanical Turk (AMT) platform, with each participant assigned a random subset to evaluate. All participants were compensated according to the platform's guidelines, ensuring a diverse pool of evaluators and promoting reliable, unbiased annotations.

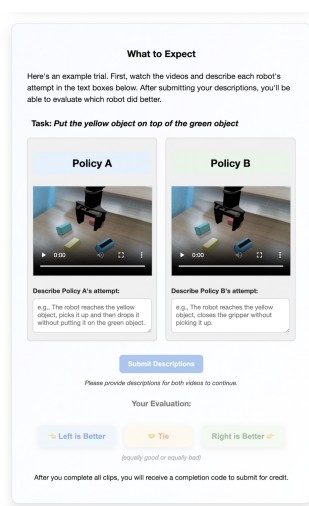

Figure 20: Human evaluation interface. Participants viewed two policies side-by-side and selected the one that performed better or if it was a tie.

## GLOBAL RANKING FROM PAIRWISE PREFERENCES

Let $\Pi = \{\pi_1, \ldots, \pi_N\}$ denote the set of policies, and let the dataset of pairwise preferences be

$$\mathcal{D}_p = \{(P_{\pi_A, \pi_B}, t)\},$$

where $P_{\pi_A, \pi_B} \in \{-1, 0, 1\}$ indicates a preference for $\pi_A$ over $\pi_B$ (1), a tie (0), or $\pi_B$ over $\pi_A$ (-1), and $t$ identifies the task on which the comparison was made.

We model the probability of each comparison using the **Bradley–Terry (BT) model**. Let $\theta_i > 0$ be the latent ability, or score, of policy $\pi_i$. For any pair of policies $(\pi_i, \pi_j)$, the probability that $\pi_i$ is preferred to $\pi_j$ is given by:

$$P(\pi_i \succ \pi_j) = \frac{\theta_i}{\theta_i + \theta_j},$$

$$P(\pi_j \succ \pi_i) = \frac{\theta_j}{\theta_i + \theta_j}.$$

**Maximum Likelihood Estimation.** The parameters $\theta = \{\theta_1, \ldots, \theta_N\}$ are estimated by maximizing the likelihood of the observed human preferences:

$$\mathcal{L}(\theta) = \sum_{(i,j) \in \mathcal{D}_p} \left[ \mathbf{1}[P_{i,j} = 1] \log P(\pi_i \succ \pi_j) + \mathbf{1}[P_{i,j} = -1] \log P(\pi_j \succ \pi_i) \right].$$

This objective is concave in the natural log-parameterization $\log \theta_i$, and we can find the maximum efficiently using Newton–Raphson optimization algorithm. We can use other iterative algorithms as well.

**Confidence Intervals and Sandwich Variance.** To quantify uncertainty in the estimated abilities, we can compute standard errors using a robust (sandwich) estimator. Let $\beta_i = \log \theta_i$ denote the log-ability of policy $\pi_i$, and $\hat{\beta}$ be the MLE.

The score function is:

$$U(\beta) = \frac{\partial \mathcal{L}}{\partial \beta} = \sum_{(i,j) \in \mathcal{D}_p} \Big( \mathbf{1}[P_{i,j} = 1] - P(\pi_i \succ \pi_j) \Big) \mathbf{x}_{ij},$$

where $\mathbf{x}_{ij}$ is a vector with $+1$ at position $i$, $-1$ at $j$, and $0$ elsewhere.

The observed Fisher information (negative Hessian) is:

$$\mathbf{H} = -\frac{\partial^2 \mathcal{L}}{\partial \beta \, \partial \beta^\top} = \sum_{(i,j) \in \mathcal{D}_p} P(\pi_i \succ \pi_j)\big(1 - P(\pi_i \succ \pi_j)\big) \mathbf{x}_{ij}\mathbf{x}_{ij}^\top.$$

The outer product of the score vectors is:

$$\mathbf{S} = \sum_{(i,j) \in \mathcal{D}_p} U_{ij}(\hat{\beta})U_{ij}(\hat{\beta})^\top, \quad U_{ij}(\hat{\beta}) = \Big( \mathbf{1}[P_{i,j} = 1] - P(\pi_i \succ \pi_j; \hat{\beta}) \Big) \mathbf{x}_{ij}.$$

The robust (sandwich) covariance estimator is then given by:

$$\mathbf{V}_{\text{sandwich}} = \mathbf{H}^{-1}\mathbf{S}\,\mathbf{H}^{-1}.$$

Finally, to center the scores and remove the global offset (since adding a constant to all $\beta_i$ does not change relative preferences), we compute:

$$\tilde{\mathbf{V}} = \mathbf{A}\,\mathbf{V}_{\text{sandwich}}\,\mathbf{A}^\top, \quad \mathbf{A} = \mathbf{I}_N - \frac{1}{N}\mathbf{1}_N\mathbf{1}_N^\top.$$

The $(1 - \alpha)$ confidence interval for each policy score is:

$$\mathrm{CI}_i = \hat{\beta}_i \pm z_{1-\alpha/2}\sqrt{\tilde{V}_{ii}},$$

where $z_{1-\alpha/2}$ is the standard normal quantile.

**Global Ranking.** A global ranking $\mathcal{R}$ over policies can be obtained by ordering the estimated ability scores $\hat{\theta}_i$ (or centered log-scores $\hat{\beta}_i$). Approximate rankings can also account for uncertainty: if the confidence intervals of two policies do not overlap, one can confidently assert that one is preferred over the other according to human evaluations.

## K  MEASURING CORRELATION BETWEEN VLM AND HUMAN SCORES

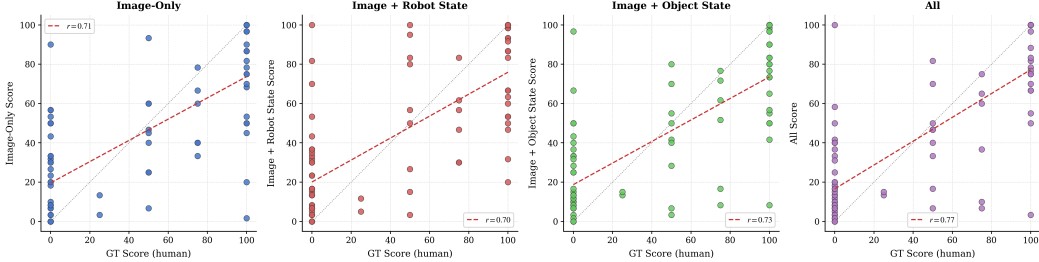

Figure 21: Measuring Correlation between VLM and Human scores.

Figure 21 shows the correlation between VLM-derived scores and human ground-truth scores under four input settings: images only, robot state only, object state only, and combined robot and object states. Each point represents a trajectory, with the dashed red line indicating the regression fit and the gray diagonal representing perfect agreement. In all cases, VLM scores correlate positively with human evaluations, with stronger alignment when structured state information is provided, particularly when both robot and object states are available.

## L  ETHICS STATEMENT

This work complies with the ICLR Code of Ethics and established research integrity principles. No human subjects, personal, or sensitive data were used for training or evaluating policies; all human involvement in the benchmarking framework is limited to online preference annotation of video content from robot simulation, with participants recruited and compensated through recognized platforms according to their guidelines. Annotator tasks were designed to be non-intrusive, requiring only high-level performance comparisons in a double-blind interface, and a qualification quiz was used to promote reliable, unbiased evaluations. Research artifacts, algorithms, and datasets are used in compliance with their licenses, and no intellectual property or privacy concerns are raised by the experimental protocol. The study sought to anticipate and mitigate unfair outcomes and measured bias and robustness across diverse input sources, including systematic stress-tests for failure modes. Limitations concerning simulation realism and generalization beyond the studied benchmarks are acknowledged, as discussed in the main text and the limitations section. The full evaluation framework and results will be released for broad, equitable access by the research community.

## M  REPRODUCIBILITY STATEMENT

Significant effort has been devoted to ensuring that all results in this paper are reproducible. Details of the reality-to-simulation translation pipeline, environment and object generation, evaluation methodology (including VLM progress scoring, human preference infrastructure, and global policy ranking algorithm), as well as hyperparameter settings, are provided in the main text and in Appendices B–G. The codebase for environment construction, simulation, and scoring, along with evaluation logs and benchmark scenarios, will be anonymously released as supplementary materials. Source datasets, simulation assets, and parameter optimization procedures are fully documented in the Appendix and will be indexed for direct replication. Further, systematic perturbation experiments and robustness analyses can be reproduced using the public release. The aim is to enable independent verification and extension of all primary claims and experimental findings using the open-source pipeline described in the manuscript.

## N  LLM USAGE STATEMENT

In the development of this manuscript, Large Language Models (LLMs) were used as general-purpose assistive tools for grammar checking, rewording, preparation of summary paragraphs, and technical editing. No sections of the manuscript, experimental results, or research claims were generated solely by an LLM. All original research content, analyses, and experiment design were authored by the stated researchers. When LLMs contributed to polishing language or format, the outputs were reviewed, edited, and verified for accuracy to ensure compliance with the ICLR Code of Ethics, integrity, and originality. The authors take full responsibility for all content presented here and confirm that no LLM-generated text constitutes plagiarism, fabrication, or any form of scientific misconduct. LLMs are not considered authors and have not contributed to research ideation, methodology, or core scientific findings