# OpenReview forum: "RobotArena $\infty$: Scalable Robot Benchmarking via Real-to-Sim Translation"
_ICLR.cc/2026/Conference — ICLR 2026 Poster_

### Official Review · Reviewer_dV9J · 2025-10-28

**Soundness:** 3
**Presentation:** 4
**Contribution:** 4
**Rating:** 8
**Confidence:** 4

**Summary:**

This paper provides a fairly comprehensive benchmarking system for evaluating VLA models for robotics. The contributions of this work are:

- A benchmarking protocol for robotics in simulation. This includes VLM-based task evaluation and preference modeling.
- An automated video to simulation pipeline.
- Extensive evaluation of many VLA models.
- Evaluation results across in-distribution and out-of-distribution cases.

**Strengths:**

- This work provides a powerful framework for the robotics community to develop on top of. This kind of work can really make an impact on scaling the data necessary for VLA-style models to succeed.
- The breadth of this work is quite remarkable, from automating video-to-simulation to evaluation across many of the open-source VLAs to extensive evaluation results.
- This work fills in significant gaps in an end-to-end pipeline that would make robot training closer to the approaches taken for training LLMs.
- The presentation of this work is well done, with thoughtful and descriptive figures and clearly robust evaluation.

**Weaknesses:**

- Overall, the work is quite strong, but there could be some clarifications that could strengthen the claims in the paper.
- On the claims of comprehensive evaluation, the authors should comment on the evaluation of diverse language instructions from users, as this is a reasonable failure mode of VLAs (e.g. https://arxiv.org/pdf/2411.18676)
- The authors should make it clear what aspects of the pipeline are truly new vs. aggregations of existing approaches. For instance, how original is the approach for real-to-sim? The work doesn’t need to be completely novel, but it would have been nice to see the authors comment on how this approach draws from prior work.

**Questions:**

- There doesn’t seem to be a discussion on the evaluation of diverse language instructions from users in this work. Can the authors comment on this?
- Related works:
    - Could the authors discuss other works that similarly provide a real to sim pipeline, such as https://arxiv.org/pdf/2403.03949? I think it would be helpful to understand how much of this work is really an aggregation of existing approaches vs. novel contributions that the reader can take away.

---

> ### Author Response · Authors · 2025-11-22
> **Response to Reviewer dV9J (1/1)**
>
> We thank the reviewer for their positive assessment of our work and are pleased that they found the contribution significant and broadly valuable to the robotics community. We appreciate the recognition that our framework “provides a powerful foundation for the community to develop on top of” and that “the breadth of this work is quite remarkable, from automating video-to-simulation to evaluating many open-source VLAs.” We are especially encouraged by the acknowledgment that our pipeline “fills in significant gaps in an end-to-end system that brings robot training closer to the methodologies used for training LLMs,” as well as the comments highlighting the clarity of our presentation and the robustness of our evaluation.
>
> Below, we address the reviewer's questions and comments in detail.
>
> ---
>
> > [R1] **"The authors should make it clear what aspects of the pipeline are truly new vs. aggregations of existing approaches. For instance, how original is the approach for real-to-sim? The work doesn’t need to be completely novel, but it would have been nice to see the authors comment on how this approach draws from prior work. Could the authors discuss other works that similarly provide a real to sim pipeline, such as [https://arxiv.org/pdf/2403.03949](https://arxiv.org/pdf/2403.03949)?"**
>
> We appreciate the suggestion to contextualize our method relative to existing real-to-sim systems. We will add a dedicated related work discussion. Briefly:
>
> * Phone2Proc[1], Scalable Real2Sim[2], and Re3Sim[3] rely on multi-view captures, curated object libraries, or fiducial markers, making them impractical for typical single-view robot datasets (e.g., Bridge, DROID). Specifically the work of [https://arxiv.org/pdf/2403.03949](https://arxiv.org/pdf/2403.03949) requires a video of a static scene as input, which our framework only requires a single RGB image.
> * RialTo[4] requires human-in-the-loop segmentation or assumes access to the robot for capturing calibration trajectories.
> * RoboTwin[5] focuses on object-centric reconstruction, whereas our benchmark requires full-scene composition and controllable perturbations.
>
> In contrast to the previous works, RobotArena performs fully automated, single-view reconstruction from a single static-camera demonstration, without assuming multi-view data, robot participation during reconstruction, or any manual segmentation. This design choice is essential for scaling to thousands of real demonstrations and for enabling systematic generalization testing of VLAs. RobotArena introduces an end-to-end real-to-sim pipeline out of existing VLMs and foundation image-to-3D mesh models. Our contribution is not in inventing each module from scratch, but in combining them into the first fully automated, single-view reconstruction pipeline. Submodules closely related to our real-to-sim components have begun to appear in recent arXiv papers e.g. [6] (Sept 26; ICLR submission deadline was Sep 24), such as the object segmentation and 3D object reconstruction. Our calibration module that estimates the robot-camera transformation fills a practical need not addressed in prior or concurrent literature.
>
> * [1] Phone2Proc: Bringing Robust Robots Into Our Chaotic World
> * [2] Scalable Real2Sim: Physics-Aware Asset Generation Via Robotic Pick-and-Place Setups
> * [3] Re3Sim: Generating High-Fidelity Simulation Data via 3D-Photorealistic Real-to-Sim for Robotic Manipulation
> * [4] Reconciling Reality through Simulation: A Real-to-Sim-to-Real Approach for Robust Manipulation
> * [5] RoboTwin: Dual-Arm Robot Benchmark with Generative Digital Twins
> * [6] Robot Learning from Any Images, Sept 26 on arxiv
>
> ---
>
> > [R2] **"There doesn’t seem to be a discussion on the evaluation of diverse language instructions from users in this work. Can the authors comment on this?”**
>
> We agree that language variation is a critical failure mode for VLAs. In the revision, we will include results on reconstructed environments using paraphrased and VLM-generated instruction variants (“similar task, different language”). RobotArena is explicitly designed to expose such weaknesses by decoupling linguistic shifts from visual and physical ones.

---

> ### Comment · Reviewer_dV9J · 2025-11-27
>
> Thank you for the clarifications. I will maintain my positive score.

---

### Official Review · Reviewer_f9Cq · 2025-10-31

**Soundness:** 2
**Presentation:** 3
**Contribution:** 2
**Rating:** 4
**Confidence:** 4

**Summary:**

This paper presents a scalable benchmarking framework for evaluating real-world–trained vision-language-action (VLA) robot policies. Traditional real-world testing is costly, slow, and unsafe, while existing simulation benchmarks lack the realism to assess models grounded in real demonstrations. The proposed method bridges this gap by automatically converting real robot videos into simulated digital twins using advances in vision-language modeling, 2D-to-3D generation, and differentiable rendering. Within these simulations, robot policies are evaluated through both automated vision-language-model scoring and large-scale human preference judgments collected online. The framework also introduces systematic environment perturbations to test robustness, enabling reproducible, human-aligned, and scalable evaluation of general-purpose robot manipulation agents.

**Strengths:**

1. The paper introduces a comprehensive and automated pipeline that seamlessly bridges real-world robot data and simulation, enabling scalable and reproducible evaluation of vision-language-action models.
2. It conducts the largest cross-lab evaluation to date, providing unprecedented insights into the generalization capabilities and limitations of current generalist robot policies under diverse distribution shifts.
3. The paper is well-written, presenting a complex technical system with conceptual clarity and a logical narrative that is easy to follow.

**Weaknesses:**

My main concerns lie in the proposed real2sim pipeline, including:

1. The 3D assets are primarily from some 3D generation models, which may produce meshes with different shape with real objects, or have implausible collisions. Moreover, the physical parameters are given by some LLMs, which can also result in some implausible physical movements in simulation.
2. Inpainting background makes the camera viewpoint in this evaluation process remain the same as, or close to, that in the original video. However, we often require other camera or multiple cameras for policy inference, e.g., the wrist/first/third view cameras.
3. The pipeline requires system identification of robots, which can cause much extra data collection and human efforts.

**Questions:**

1. I am curious about the evaluation results of the pi series VLA models on the proposed benchmark.
2. Can you provide more detailed information of your sim-real alignment experiments? For example, what are the controllers and control frequencies of the simulation and real environments, since controller type and frequency can sometimes cause different motions in simulation and real world?

---

> ### Author Response · Authors · 2025-11-22
> **Response to Reviewer f9Cq (1/2)**
>
> We are highly encouraged that you recognize the significant contribution of our novel and automated pipeline, which “seamlessly bridges real-world robot data and simulation, enabling scalable and reproducible evaluation of vision-language-action models.” We also appreciate your acknowledgment of our extensive empirical work, noting that we “conduct the largest evaluation to date, providing unprecedented insights into the generalization capabilities and limitations of current generalist robot policies under diverse distribution shifts.” Finally, your positive feedback on the clarity of our presentation—that the “paper is well-written, presenting a complex technical system with conceptual clarity and a logical narrative that is easy to follow”—validates our effort to make this complex framework accessible.
>
> Below, we address your questions and comments in detail.
>
> ---
>
> > [R1] **"The 3D assets are primarily from some 3D generation models, which may produce meshes with different shapes with real objects, or have implausible collisions."**
>
> We acknowledge your concern that 3D assets generated by models may introduce geometric inaccuracies, potentially leading to implausible shapes or simulation collisions. To enhance robustness, we automatically convexify mesh collision geometries using CoACD [1] prior to simulation. Our use of CoACD ensures that even if the visual mesh has minor artifacts, the physical collision mesh remains stable. While we plan to explore asset retrieval in the future, we currently favor image-to-mesh generation because it preserves the original texture and shape more faithfully.
>
> The generated 3D assets are sufficiently accurate to reproduce training-set trajectories, confirming that they meet the fidelity requirements for our benchmark. Moreover, this is a rapidly advancing area—recent developments such as Meta’s sam3D model highlight the pace of progress. We expect that the fidelity of generated assets will continue to improve, and our current experiments indicate that they are already of adequate quality for our purposes. We will clarify these points in the revision.
>
> *[1] Approximate Convex Decomposition for 3D Meshes with Collision-Aware Concavity and Tree Search*
>
> ---
>
> > [R2] **"The physical parameters are given by some LLMs, which can also result in some implausible physical movements in simulation."**
>
> We have generated over 500 assets and so far have not observed any implausible physical movements in simulation. The LLMs can generate reasonable physical parameters for a variety of objects, and in fact this method is used in many previous approaches to infer physical properties for object and object image crops.
>
> ---
>
> > [R3] **"Inpainting background makes the camera viewpoint in this evaluation process remain the same as, or close to, that in the original video. However, we often require other cameras or multiple cameras for policy inference, e.g., the wrist/first/third view cameras."**
> >
> > [R4] **"I am curious about the evaluation results of the pi series VLA models on the proposed benchmark."**
>
> Thank you for raising this point. At present, RobotArena does not support multi-view camera configurations. Policies such as the Pi-series rely on wrist-camera inputs, and therefore may not achieve their full performance when limited to the head camera alone. To address your concern, we will evaluate Pi-0 and any other VLAs that allow it by disabling the wrist-camera input, following each model’s recommended procedure for handling missing observations. We will report these results as soon as they are available.
>
> We agree that wrist-mounted RGB cameras constitute an important modality for VLAs. We are actively working toward upgrading our evaluation environments to support full 3D reconstruction so that wrist-camera observations can be incorporated, with the use of video generative models and 3D scene reconstructions techniques in order to obtain a complete 3D world scene from a single image. This allows us to render accurate wrist-camera inputs even when the wrist is far from the original viewpoint. We have conducted preliminary experiments to validate the feasibility of this extension and the example can be found in the website ([https://submitsquirtel.github.io/#wrist-multiview-cameras](https://submitsquirtel.github.io/#wrist-multiview-cameras)).
>
> We believe that the current version of the benchmark is still highly valuable for the community. It already provides a significantly richer and more realistic evaluation setting than SIMPLER, which is the standard currently used by VLAs, despite covering only a handful of environments, three in total.

---

> > ### Comment · Reviewer_f9Cq · 2025-11-27
> >
> > Thank you for your detailed replies! Since all my concerns have been resolved, I have decided to raise my score to 6, despite the limitation that it only works for a fixed camera view.

---

> ### Author Response · Authors · 2025-11-22
> **Response to Reviewer f9Cq (2/2)**
>
> > [R5] **"The pipeline requires system identification of robots, which can cause much extra data collection and human efforts."**
>
> System identification is performed once per robot per dataset, not per demonstration. After this single calibration step, the same parameters are reused across all demonstrations without additional human effort. For datasets that provide calibration metadata, this step can be automated entirely. We will make this clearer in the paper.
>
> ---
>
> > [R6] **"Can you provide more detailed information of your sim-real alignment experiments? For example, what are the controllers and control frequencies of the simulation and real environments, since controller type and frequency can sometimes cause different motions in simulation and real world?"**
>
> Thank you for requesting these technical details. Our sim–real alignment experiment uses:
>
> * **Simulation timestep:** 0.005 s
> * **Controller:** PD controller with the gains obtained by the system identification
> * **Matching strategy:** We replay real-world joint trajectories and end-effector targets inside simulation and compare the resulting motion traces.
>
> Here are the Errors for trajectory matching:
>
> **Average Trajectory Errors**
>
> | System ID | Pos Error | Rot Error | Combined Error |
> | :--- | :--- | :--- | :--- |
> | **Without** | 0.0330 | 0.1127 | 0.1457 |
> | **With** | 0.0234 | 0.0241 | 0.0475 |
>
> This ensures that discrepancies arise from VLA behavior, not controller mismatch. We will add explicit controller specifications to the appendix.

---

### Official Review · Reviewer_kdpq · 2025-11-01

**Soundness:** 3
**Presentation:** 3
**Contribution:** 3
**Rating:** 4
**Confidence:** 2

**Summary:**

In this work, the authors introduce a real-to-sim translation framework named RobotArena to tackle the challenges in real-world robot policy benchmarking with the following two aspects: 1) an automated pipeline for converting real-world videos into simulated environments via VLM-based scene understanding and differentiable rendering. 2) an evaluation protocol combining VLM-based progress scoring and crowdsourced human preference feedback. The authors conduct extensive experimental results on hundreds of scenarios to demonstrate the effectiveness of proposed framework on VLA-based policies.

**Strengths:**

1. The automated real-to-sim translation pipeline that the paper introduces is innovative.
2. The hybrid assessment method integrating VLM-based scoring with human preference feedback is comprehensive.

**Weaknesses:**

1. The simulation environment cannot accurately reproduce fine-grained physical interactions (e.g., plug insertion, deformable object manipulation), limiting evaluation fidelity for precision tasks.
2. The multi-stage pipeline may accumulate errors, but the paper lacks quantitative analysis of error propagation across stages.
3. The benchmark primarily focuses on static-camera, table-top manipulation tasks from datasets like Bridge and DROID, lacking coverage of dynamic scenarios, mobile navigation, or multi-object interactions.

**Questions:**

Regarding the real-to-sim translation pipeline design:
1. Although the fully automated reality-to-simulation translation pipeline is novel, could the authors provide more details on how error propagation across these stages is quantified? For instance, how do inaccuracies in VLM-based segmentation or 3D mesh generation impact the final simulation fidelity?

Regarding the evaluation methodology and potential biases:
1. The hybrid evaluation combining VLM scoring and human preferences is compelling. Yet, how do the authors mitigate biases inherent in both methods?

Regarding the experimental design and generalization claims:
1. The large-scale experiments reveal significant cross-dataset performance drops. Could the authors discuss whether the observed failures are due to visual domain shifts or task complexity differences?
2. How might the benchmark be extended to include dynamic scenarios to better assess true generalization?

---

> ### Author Response · Authors · 2025-11-22
> **Response to Reviewer kdpq (1/3)**
>
> We are grateful to the reviewer for recognizing the novelty and value of our framework. We appreciate the acknowledgment that "The automated real-to-sim translation pipeline that the paper introduces is innovative," and that the hybrid evaluation method is "comprehensive."
>
> Below, we address the reviewer's questions and comments in detail.
>
> ---
>
> > [R1] **"The simulation environment cannot accurately reproduce fine-grained physical interactions, limiting evaluation fidelity for precision tasks."**
>
> We agree that achieving high-fidelity simulation for fine-grained interactions—such as plug insertion, tight clearances, or deformable object manipulation—remains challenging. However, our framework is explicitly designed to be extensible with respect to the underlying physics solver.
>
> Our backend simulator, Genesis, supports the Incremental Potential Contact (IPC) solver [1], which provides intersection-free and inversion-free contact handling at all time steps. This makes IPC particularly well suited for accurate and stable simulation of tight contacts and high-precision manipulation.
>
> RobotArena can therefore be readily extended to evaluate precision tasks by enabling the IPC solver within Genesis. As VLAs advance and become capable of reliable, high-precision behaviors, our benchmark infrastructure will be equipped to evaluate them faithfully.
>
> In addition, existing libraries of high-precision assets—such as those used for bolt-screwing, nut-tightening, and other contact-sensitive tasks [2]—can be directly imported into our environments, further enhancing fidelity for precision evaluation.
>
> * [1] Incremental Potential Contact: Intersection- and Inversion-free Large Deformation Dynamics
> * [2] Factory: Fast Contact for Robotic Assembly
>
> ---
>
> > [R2] **"The paper should provide more details on how error propagation across the stages is quantified."**
>
> We thank the reviewer for the question. The goal of our real-to-sim pipeline is to generate high-quality evaluation arenas; it does not need to succeed on every video, only on a sufficient number to populate a robust benchmark. In practice, we find the pipeline to be highly reliable, achieving over 85% end-to-end success in converting real demonstrations into simulation-ready environments across both Bridge and DROID.
>
> To reduce and isolate error propagation, the pipeline is deliberately modular, allowing each stage (segmentation → 3D reconstruction → mesh refinement → object placement → physics assignment) to be independently validated, improved, or replaced. While a full quantitative analysis of cross-stage error propagation is beyond the scope of this submission, we offer several clarifications:
>
> * **Segmentation inaccuracies:** We employ redundant VLM and vision-based segmentation backbones and select the model with the most consistent performance across diverse tasks, reducing single-model failure modes.
> * **3D reconstruction errors:** Mesh imperfections are corrected using geometric priors (watertightness, manifoldness, and approximate scale constraints). In our experiments, remaining imperfections rarely hinder downstream VLA performance in tabletop settings.
> * **Robustness improvements:** The pipeline naturally accommodates ensembling or quality-based selection—for example, running multiple reconstruction models in parallel and choosing outputs based on geometric consistency checks. We will clarify this capability in the final version.
>
> Finally, a key advantage of our design is long-term upgradability: each module can be replaced with stronger models as real-to-sim technologies improve, continually reducing error and improving benchmark fidelity.

---

> ### Author Response · Authors · 2025-11-22
> **Response to Reviewer kdpq (2/3)**
>
> > [R3] **"The benchmark primarily focuses on static-camera, table-top manipulation tasks, lacking coverage of dynamic scenarios, mobile navigation, or multi-object interactions."**
>
> We note that there already exist numerous benchmarks for robot navigation. In contrast, there is no large-scale benchmark that can evaluate manipulation VLAs trained with real world demonstrations, which is why we chose to focus on manipulation. We acknowledge that our current version emphasizes static-camera and tabletop tasks. This reflects both the present maturity of VLAs and the constraints of widely used real-world datasets (e.g., Bridge, DROID), many of which already include multi-object interactions. We would like to note that our pipeline is not intrinsically limited to flat, rigid tabletop surfaces. It is fully capable of reconstructing scenes with structures like shelves or containers, allowing for tasks such as "put cups on the shelf."
>
> We agree that wrist-mounted RGB cameras constitute an important modality for VLAs. We are actively working toward upgrading our evaluation environments to support full 3D reconstruction so that wrist-camera observations can be incorporated, with the use of video generative models and 3D scene reconstructions techniques in order to obtain a complete 3D world scene from a single image.  This allows us to render accurate wrist-camera inputs even when the wrist is far from the original viewpoint.  We have conducted preliminary experiments to validate the feasibility of this extension and the example can be found in the website ([https://submitsquirtel.github.io/#wrist-multiview-cameras](https://submitsquirtel.github.io/#wrist-multiview-cameras)).
>
> Looking forward, we plan to generate scene environments, assets, and tasks on the fly, leveraging recent agent-based frameworks for text-to-simulation generation [1]. These developments will enable longer-horizon tasks, dynamic scenarios, and richer multi-object interactions, addressing the reviewer’s concerns in future iterations of the benchmark.
>
> While we acknowledge the limitation noted by the reviewer, we believe that the current version of the benchmark is still highly valuable for the community. It already provides a significantly richer and more realistic evaluation setting than SIMPLER, which is the standard currently used by VLAs, despite covering only a handful of environments.
>
> * [1] SCENEWEAVER- All-in-One 3D Scene Synthesis with an Extensible and Self-Reflective Agent
>
> ---
>
> > [R4] **"The authors must discuss how they mitigate biases inherent in both VLM scoring and human preference evaluation."**
>
> We thank the reviewer for raising this important point. Our VLM-based task progression builds on prior work showing that in-context VLM value estimation provides a robust and reproducible evaluation signal. Nevertheless, we take several steps to mitigate biases in both VLM scoring and human preference assessments.
>
> **Mitigating bias in human evaluations:**
>
> * **Double-blind preference interface:** Annotators are not shown the identity or source of the VLA being evaluated, preventing model-recognition or expectation-driven bias.
> * **Justification requirement:** Annotators provide a brief reasoning chain for each preference decision, which discourages superficial or stylistic judgments and encourages deliberative evaluation.
> * **Redundant annotation and aggregation:** Multiple annotators evaluate each comparison, and we aggregate preferences across individuals to reduce variance and individual annotator bias. We additionally filter out unreliable annotations: if an annotator fails embedded quality checks, all ratings from that annotator are discarded.
>
> Importantly, RobotArena is designed to intentionally capture certain human biases—such as preferences over speed, precision, smoothness, safety, and other nuanced aspects of task execution—because these reflect how users genuinely want tasks to be performed. Our framework aims to mitigate unwanted evaluative biases while still faithfully preserving the human factors that matter for robot behavior.

---

> ### Author Response · Authors · 2025-11-22
> **Response to Reviewer kdpq (3/3)**
>
> > [R5] **"The authors should discuss whether observed failures are due to visual domain shifts or task complexity differences."**
>
> The performance drops strongly correlate with visual domain shifts—including differences in lighting, camera intrinsics, clutter, object appearance, and background statistics—as well as shifts in low-level action distributions across datasets. In our experiments, this effect is pronounced because the evaluated VLAs are trained only on Bridgev2; hence, changing to a dataset with different visual distributions already constitutes a significant out-of-distribution shift. This demonstrates that VLAs fail both under within-dataset perturbations and cross-dataset visual changes, indicating that current perception modules remain far from open-world generalization. RobotArena’s design explicitly isolates these axes through controlled perturbations, allowing us to determine which factors most strongly influence these failure modes.
>
> ---
>
> > [R6] **"The authors must discuss how the benchmark might be extended to include dynamic scenarios."**
>
> We appreciate the suggestion toward dynamic tasks. While current VLA models are still mastering quasi-static manipulation, RobotArena's design makes extending to dynamic scenarios straightforward:
>
> * **Dynamic Task Construction:** We can easily extend our task definitions to include moving rigid bodies, such as tasks involving catching, intercepting, or pursuing moving targets.
> * **Solver Compatibility:** The underlying physics simulator (Genesis) can handle these dynamic interactions and velocity-based controls.

---

> > ### Comment · Reviewer_kdpq · 2025-11-27
> >
> > Thank you for the detailed response. Since the authors addressed my concerns, I have decided to raise the score to 6.

---

### Official Review · Reviewer_i6zo · 2025-11-01

**Soundness:** 3
**Presentation:** 2
**Contribution:** 4
**Rating:** 6
**Confidence:** 4

**Summary:**

This paper introduces a scalable framework for evaluating VLA policies in robotics. It automatically converts real-world robot demonstration videos into simulated environments using advances in VLMs, 2D-to-3D generative modeling, and differentiable rendering. The benchmark assesses robot policies through automated VLM-based scoring and human preference judgments, enabling large-scale, reproducible evaluations without manual setup. Experiments reveal limited cross-dataset generalization and robustness in current models, while confirming consistent performance rankings.

**Strengths:**

- The pipeline that transforms real-world robot demonstration videos into simulated environments is very nice and generally useful.
- The study conducts, according to the authors, the most extensive evaluation of generalist robot policies to date.

**Weaknesses:**

Real-to-sim pipeline is very nice. But for example, how is it better than testing VLAs on a bunch of different simulation environments? That would also make sure to include some out-of-distribution domains, wouldn't it? And why is this work framed as a policy evaluation work? It looks like a real-to-sim method, and it deserves credit for that contribution (more for it than for policy evaluation, because real-to-sim is more general).

I think the paper has a good potential, but the following improvements (or addressing these comments) would make it much stronger:
- Wrist camera is a common input, but the current work is missing them as acknowledged by the paper.
- Reliance on generative models may introduce reconstruction artifacts and physics inaccuracies.
- Benchmark scope limited to manipulation tasks: no locomotion, mobile, or multi-agent settings.
- Evaluation depends on a specific VLM backbone (Gemini). This harms accessibility. It would be nice to have open-source variants that are more accessible to research labs.
- Domain perturbations are hand-designed and may not capture natural distribution shifts.

**Questions:**

Please see the first paragraph that I wrote above in the Weaknesses section.

---

> ### Author Response · Authors · 2025-11-22
> **Response to Reviewer i6zo (1/2)**
>
> Thank you for your feedback. We are pleased that you found our submission to have excellent contribution and recognized the value of our core methodology. We appreciate the acknowledgment that "The pipeline that transforms real-world robot demonstration videos into simulated environments is very nice and generally useful," and that "The study conducts... the most extensive evaluation of generalist robot policies to date."
>
> Below, we address your questions and comments in detail.
>
> ---
>
> > [R1] **"The paper should clarify how the proposed framework is better than testing VLAs on a bunch of different simulation environments."**
>
> Thank you for this question. There are indeed benchmarks for evaluating robot policies in simulation. However, most existing benchmarks train and test policies within the same domain, which—as we explain in the introduction—does not provide a meaningful evaluation for VLAs trained on real-world data. Your suggestion is to test on a collection of simulated environments. Behavior-1K [1], for example, follows this approach by generating photorealistic environments in a physics engine to evaluate robot policies.
>
> We initially explored this direction, but found that all policies exhibited very low performance. We realized that current VLAs do not yet demonstrate the level of generalization required to perform well in entirely new domains, including newly generated simulated ones. For this reason, to obtain meaningful performance signals, we designed our simulated evaluation environments to be closer to the training distribution of modern VLAs through our real-to-sim translation.
>
> That said, we plan to expand our evaluation domains to include a wider range of simulated environments and novel tasks. We expect that robot policies will continue to improve and will soon achieve the degree of generalization necessary for such broader evaluations.
>
> *[1] BEHAVIOR-1K: A Human-Centered, Embodied AI Benchmark with 1,000 Everyday Activities and Realistic Simulation*
>
> ---
>
> > [R2] **"The work should justify why it is framed as a policy evaluation work instead of a real-to-sim method."**
>
> We appreciate the reviewer’s observation that RobotArena introduces a novel real-to-sim reconstruction pipeline. Its purpose is to enable rigorous evaluation of robot policies, an area that remains significantly underexplored despite its importance. Current VLAs lack reliable benchmarks that reveal their true generalization capabilities, and addressing this gap is the central motivation of our paper.
>
> Our calibration module that estimates the robot-camera transformation fills a practical need not addressed in prior or concurrent literature. We agree that the coexistence of a new real-to-sim pipeline and a policy evaluation framework may have caused confusion, and we have revised the paper to more clearly articulate this dual contribution:
>
> 1.  A real-to-sim pipeline developed to support large-scale, realistic policy evaluation, and
> 2.  The resulting benchmark and human preference crowdsourcing that provides the first systematic assessment of VLA generalization.
>
> ---
>
> > [R3] **"The current work is missing the wrist camera as a common input."**
>
> We agree that wrist-mounted RGB cameras constitute an important modality for VLAs. We are actively working toward upgrading our evaluation environments to support full 3D reconstruction so that wrist-camera observations can be incorporated, with the use of video generative models and 3D scene reconstructions techniques in order to obtain a complete 3D world scene from a single image. This allows us to render accurate wrist-camera inputs even when the wrist is far from the original viewpoint. We have conducted preliminary experiments to validate the feasibility of this extension and the example can be found in the website ([https://submitsquirtel.github.io/#wrist-multiview-cameras](https://submitsquirtel.github.io/#wrist-multiview-cameras)).
>
> While we acknowledge the limitation noted by the reviewer, we believe that the current version of the benchmark is still highly valuable for the community. It already provides a significantly richer and more realistic evaluation setting than SIMPLER, which is the standard currently used by VLAs, despite covering only a handful of environments, three in total.

---

> ### Author Response · Authors · 2025-11-22
> **Response to Reviewer i6zo (2/2)**
>
> > [R4] **"Reliance on generative models may introduce reconstruction artifacts and physics inaccuracies."**
>
> We acknowledge your concern that 3D assets generated by models may introduce geometric inaccuracies, potentially leading to implausible shapes or simulation collisions. In practice, however, this has not posed a bottleneck for VLA evaluation. Across 500 assets generated using Hunyuan3D, we have not observed failures that negatively affect VLA performance.
>
> To further enhance robustness, we automatically convexify mesh collision geometries using CoACD [1] prior to simulation, which substantially reduces the likelihood of implausible contacts. While we plan to explore asset retrieval in the future, we currently favor image-to-mesh generation because it preserves the original texture and shape more faithfully.
>
> The generated 3D assets are sufficiently accurate to reproduce training-set trajectories, confirming that they meet the fidelity requirements for our benchmark. Moreover, this is a rapidly advancing area—recent developments such as Meta’s sam3D model highlight the pace of progress. We expect that the fidelity of generated assets will continue to improve, and our current experiments indicate that they are already of adequate quality for our purposes.
>
> *[1] Approximate Convex Decomposition for 3D Meshes with Collision-Aware Concavity and Tree Search*
>
> ---
>
> > [R5] **"The benchmark scope is limited to manipulation tasks: no locomotion, mobile, or multi-agent settings."**
>
> We acknowledge that our benchmark focuses on tabletop manipulation tasks. This reflects our initial goal of establishing a reliable and controlled benchmark for VLA evaluation. That said, we are actively working toward extending the benchmark to mobile manipulation and broader task categories, as supported by our ongoing full 3D reconstruction efforts and our work on novel simulation environments and task generation (Please see [R2] and [R3]). These developments will enable more diverse settings, including locomotion and multi-agent scenarios, in future versions of the benchmark.
>
> ---
>
> > [R6] **"Evaluation depends on a specific VLM backbone (Gemini), which harms accessibility."**
>
> We used Gemini as it represents the current state-of-the-art visual understanding, and such capability is necessary for accuracy of progress monitoring. We believe any other VLMs that have similar performance in visual understanding can serve as an alternative backbone, providing reasonable progress monitoring. Our benchmark uses VLM evaluations as a faster and cheaper way over human preferences, but similar progress evaluation scores are now collected from human users in our web interface. Last, an area of our current research is exploring automated success detector/progress evaluation that exploits the simulator's ground-truth state (such as object 3D poses and 3D locations), moving beyond image-based evaluation with VLMs for broader and cheaper accessibility.
>
> ---
>
> > [R7] **"Domain perturbations are hand-designed and may not capture natural distribution shifts."**
>
> The perturbation we chose to conduct is standard in the literature [2]. Our aim is not necessarily to mimic natural distribution shifts, but rather to systematically stress-test policies along controlled, isolated dimensions of variation (e.g., background change, color shift, and object pose change). This controlled perturbation framework allows us to isolate and identify causal weaknesses in generalization—an objective that natural, uncontrolled shifts cannot achieve. We will make this motivation clearer in the manuscript.
>
> We hope this clarifies your questions and we hope to hear your feedback. We would be very happy to provide further details during the remainder of the rebuttal period if you have additional questions or comments.
>
> *[2] Evaluating Real-World Robot Manipulation Policies in Simulation*

---

### Meta-Review · Area_Chair_xzEr · 2026-01-06

**Summary:**

The reviewers have concerns about the fidelity of Real2Sim pipeline on complex tasks, the overly simplified settings, and the potential accumulated errors. Moreover, the advantage against other similar pipelines are not discussed carefully.

**Reviewer Concerns:**

Most of the concerns are resolved during the rebuttal.

**Reviewer Scores:**

6 6 6 8

---

### Decision · Program_Chairs · 2026-01-26

Accept (Poster)